# An integrated workflow for quantitative analysis of the newly synthesized proteome

Toman Borteçen[1,2], Torsten Müller[3] & Jeroen Krijgsveld [1,3] ✉

The analysis of proteins that are newly synthesized upon a cellular perturbation can provide detailed insight into the proteomic response that is elicited by specific cues. This can be investigated by pulse-labeling of cells with clickable and stable-isotope-coded amino acids for the enrichment and mass spectrometric characterization of newly synthesized proteins (NSPs), however convoluted protocols prohibit their routine application. Here we report the optimization of multiple steps in sample preparation, mass spectrometry and data analysis, and we integrate them into a semi-automated workflow for the quantitative analysis of the newly synthesized proteome (QuaNPA). Reduced input requirements and data-independent acquisition (DIA) enable the analysis of triple-SILAC-labeled NSP samples, with enhanced throughput while featuring high quantitative accuracy. We apply QuaNPA to investigate the time-resolved cellular response to interferon-gamma (IFNg), observing rapid induction of targets 2 h after IFNg treatment. QuaNPA provides a powerful approach for large-scale investigation of NSPs to gain insight into complex cellular processes.

Cells reshape their proteome in response to external stimuli or stress, which occurs throughout the cell's lifetime, e.g., under the influence of growth factors to induce differentiation, or upon genotoxic stress to enhance cell survival or leading to cell death. To investigate these processes, proteome analysis by mass spectrometry, antibodies or aptamers typically is performed to measure differences in overall protein abundance levels between cellular conditions. On the one hand, this resolves the concern of transcriptome analysis where accumulating evidence indicates that the correlation between mRNA and protein expression is usually low[1,2]. On the other hand, observing a change in the abundance of a given protein does not indicate the mechanism and dynamics by which this has occurred. Since mRNA translation is central to rewire protein expression upon cellular perturbations and in disease[3,4], the analysis of protein synthesis is key to fill this gap, in particular, because it can reveal immediate changes in the proteome even before this becomes apparent as a change in overall protein abundance[5]. Importantly, measuring these early events, and distinguishing them from proteins exhibiting a secondary or delayed response in protein synthesis, are crucial to gain insight into the underlying mechanisms that translate a cellular perturbation into a proteomic response.

Experimentally, protein synthesis has been investigated by ribosome profiling, inferring protein translation by genome-wide sequencing of ribosome occupancy sites in mRNA (Ribo-seq). This can be combined with global RNA sequencing to obtain a measure for translation efficiency (TE) by accounting for changes in mRNA expression[6]. Yet, Ribo-seq determines protein translation only indirectly by sequencing of RNA, and it comes with a number of caveats especially when studying cellular perturbation by requiring rigorous statistical methods to faithfully calculate differences in TE[7,8], and by the need to disregard inactive ribosomes that do not contribute to translation[9].

Therefore, a number of mass spectrometry-based proteomic methods have been developed to identify newly synthesized proteins (NSPs) directly at the protein level[10,11], based on the incorporation of puromycin (or its derivatives), isotope-labeled amino acids, non-natural amino acids, or combinations thereof[12,13]. Puromycin is incorporated into nascent chains, leading to termination of protein translation and release of the truncated polypeptide which can next be

[1]German Cancer Research Center (DKFZ), Im Neuenheimer Feld 581, Heidelberg, Germany. [2]Heidelberg University, Faculty of Biosciences, Im Neuenheimer Feld 581, Heidelberg, Germany. [3]Heidelberg University, Medical Faculty, Im Neuenheimer Feld 581, Heidelberg, Germany. ✉e-mail: j.krijgsveld@dkfz.de

isolated via biotin-streptavidin[14] enrichment, click-chemistry[15–17] or anti-puromycin antibodies[18], depending on the type of puromycin-analog that is used. This can be combined with stable isotope labeling with amino acid in cell culture (SILAC) to distinguish NSPs from pre-existing or contaminating proteins by subsequent mass spectrometric analyses. Pulsed-SILAC (pSILAC) labeling alone can also be used to label and analyze full-length NSPs that have undergone natural translation termination. As a third approach, non-natural methionine analogs such as L-azidohomoalanine (AHA)[19], L-homopropargylglycine (HPG)[20] or L-azidonorleucine (ANL)[21] can be incorporated into nascent proteins by the cell's translational machinery, exploiting their bioorthogonal alkyne or azide moieties for subsequent coupling of NSPs to immobilized tags[22], to biotin-conjugates[23], phosphonate alkynes[24] or directly to clickable beads[25]. This covalent (or near-covalent in the case of streptavidin) capture of NSPs has the great advantage to allow stringent washing to remove pre-existing (i.e., non-labeled) proteins and other contaminants before digestion of NSPs off the beads and analysis by liquid chromatography coupled to mass spectrometry (LC-MS).

Several strategies have combined the use of clickable and isotope-labeled amino acids to enrich and quantify NSPs in the same experiment. This includes the use of isotope-labeled AHA for the quantification of newly synthesized AHA-containing peptides[26], or the simultaneous pulse-labeling of cells with AHA and SILAC amino acids[12,13,23,27]. An important advantage of these strategies is that the detection of isotope-labeled peptides is used as formal evidence for their assignment as NSPs, and to allow relative quantification of NSPs between conditions. These approaches have been applied to investigate proteome response in various model systems including cell culture[28–30], in T cells[23], and in mouse tissue both ex vivo[31] and in vivo[32]. In our own work, we have combined AHA and pSILAC labeling in various biological contexts, e.g., to investigate secreted proteins[27,33], to determine proteomic effects of rRNA methylation[34,35], and to identify effectors of transcriptional regulators[36], thus illustrating broad applicability. In addition, time-course analysis of macrophage activation showed that robust changes in NSPs can be detected on shorter time scales than in conventional proteome profiling or by the use of pSILAC without NSP enrichment[5].

Despite conceptual advantages, bio-orthogonal NSP enrichment approaches are limited in one or multiple ways with regard to throughput, required input amounts, manual and multistep sample preparation, and proteomic depth[26,28,29]. For instance, relatively large sample input is needed to isolate the usually small fraction of NSPs, and peptide fractionation or long LC gradients are needed prior to LC-MS to achieve sufficient proteome coverage, however leading to reduced throughput[5,14,17]. To avoid manual sample processing via extensive enrichment protocols, two automated enrichment methods have been developed recently, although they still include lengthy off-deck dephosphorylation and dialysis steps, while requiring large sample input and reporting limited proteome coverage[24,37]. Additional multiplexing with isobaric tags has been used to increase throughput but still requires offline sample fractionation and prolonged LC-MS measurement time to achieve sufficient proteomic depth[30,38,39].

Recently, mass spectrometry via data-independent acquisition (DIA) has become a powerful alternative to conventional data-dependent acquisition (DDA) methods for deep proteome profiling in single-shot analyses[40]. Although DIA is directly compatible with label-free quantification, it has been scarcely applied in combination with SILAC labeling because of challenges in data analysis, which only very few software tools are capable of processing[41–43]. Recently, a workflow called plexDIA was introduced and integrated into the DIA-NN software environment. Using plexDIA deep proteome coverage and quantitative accuracy were demonstrated for the analysis of multiplexed samples with non-isobaric labels (mTRAQ)[44].

Conceptually, plexDIA could also be applied to SILAC-labeled samples, in particular for the analysis of NSPs.

Here we addressed these various challenges in the analysis of NSPs, improving multiple steps in sample preparation, mass spectrometry and data analysis that we integrated into an efficient workflow named QuaNPA (Quantitative Newly synthesized Proteome Analysis). QuaNPA is centered around automated enrichment, clean-up and digestion of clickable NSPs on a liquid handling robot, which we facilitated by designing high-capacity magnetic alkyne agarose (MAA) beads. Through the use of MAA beads, the required protein input was significantly reduced, allowing cells to be grown in 6-well plates, thus making large-scale cell culture experiments a manageable task. Finally, we established that plexDIA enabled the analysis of triple-SILAC-labeled NSP samples by DIA, with quantitative accuracy equivalent to DDA. Importantly, this can be performed in single LC-MS runs, obviating the need for offline peptide fractionation, and thereby achieving a significant increase in throughput. We demonstrate the utility of QuaNPA in a time-series experiment to investigate the cellular response to interferon-gamma (IFNg), showing expression of known and previously unrecognized IFNg response proteins among NSPs at distinct time points, and as early as 2 h after the addition of IFNg. Collectively, QuaNPA presents a unified approach for systematic NSP analyses across multiple cellular conditions, to understand perturbation-induced proteome responses.

## Results

### Developing an improved workflow for proteome-wide analysis of newly synthesized proteins

We developed a complete workflow for quantitative newly synthesized proteome analysis (coined QuaNPA), that integrates metabolic labeling of cells, cell lysis, enrichment of newly synthesized proteins, sample clean-up, LC-MS measurement, and data analysis. Here, we have optimized multiple aspects of these individual steps and adapted them for automated processing, with a focus on reducing sample input, and on increasing proteomic depth and sample throughput (Fig. 1).

### Optimizing the automated enrichment of newly synthesized proteins with magnetic alkyne agarose beads

At the core of the QuaNPA workflow is the metabolic labeling of cells, to incorporate AHA and SILAC amino acids for the capture and quantification, respectively, of newly synthesized proteins (NSPs). To enable efficient and automated enrichment of AHA-containing NSPs, magnetic beads with high density of terminal alkyne groups are beneficial. Since commercially available magnetic alkyne beads lack capacity (30–50 nmol/mg beads), while regular alkyne agarose beads have a high capacity (10–20 μmol/mL resin) but lack magnetic properties to permit automation, we combined the benefit of both by producing magnetic alkyne agarose (MAA) beads by coupling epoxy-activated magnetic agarose with propargylamine. This can be performed in a one-step reaction to generate a large batch for multiple enrichment experiments (Fig. 2). Next, we established a protocol on a Bravo liquid handling system to perform enrichment of newly synthesized proteins via click-chemistry in a semi-automated fashion, using MAA beads and a magnetic rack. The automated protocol is carried out using a 96-well PCR plate, enabling the parallel processing of 8–96 samples including dispensing of reagents, click-based coupling of NSPs to MAA beads, and stringent washing steps, all in a volume of <200 μL per sample. Next, enriched NSPs are digested off the beads by trypsin, followed by peptide purification using the autoSP3 protocol[45]. To enable the dilution of the tryptic peptides to >95% acetonitrile, samples were lyophilized after digestion prior to the addition of magnetic SP3-beads and acetonitrile. The automated SP3 peptide clean-up protocol was run on the same Bravo liquid handling platform after exchange of the "reagent" plate to a new 96-well plate

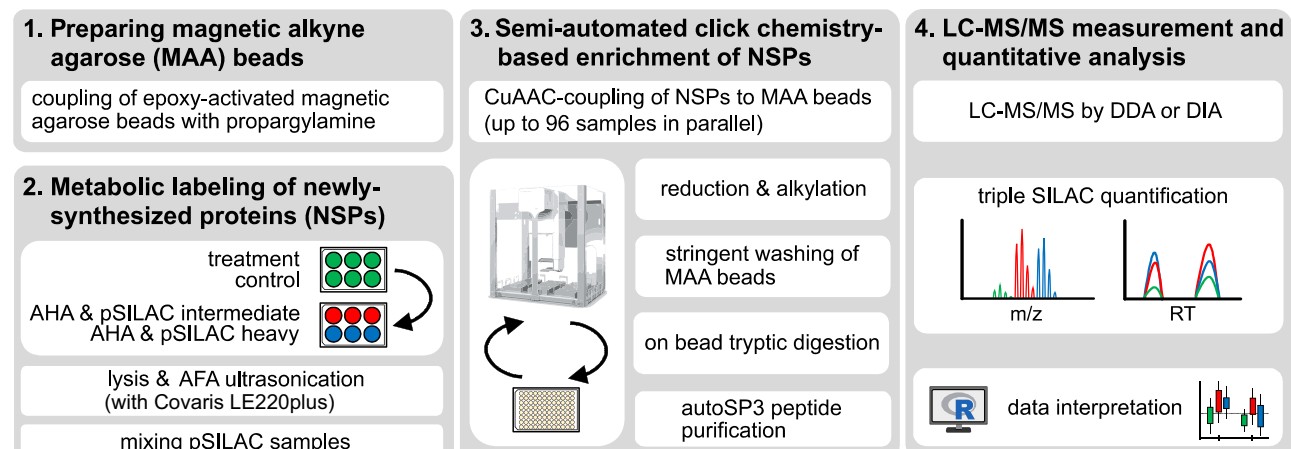

**1. Preparing magnetic alkyne agarose (MAA) beads**

coupling of epoxy-activated magnetic agarose beads with propargylamine

**2. Metabolic labeling of newly-synthesized proteins (NSPs)**

treatment control
AHA & pSILAC intermediate
AHA & pSILAC heavy

lysis & AFA ultrasonication (with Covaris LE220plus)

mixing pSILAC samples

**3. Semi-automated click chemistry-based enrichment of NSPs**

CuAAC-coupling of NSPs to MAA beads (up to 96 samples in parallel)

reduction & alkylation

stringent washing of MAA beads

on bead tryptic digestion

autoSP3 peptide purification

**4. LC-MS/MS measurement and quantitative analysis**

LC-MS/MS by DDA or DIA

triple SILAC quantification

m/z          RT

data interpretation

**Fig. 1 | Schematic representation of the workflow for quantitative newly synthesized proteome analysis with automated sample preparation (QuaNPA).** The workflow consists of four main steps. (1) In preparation for the enrichment of newly synthesized proteins (NSP), epoxy-activated magnetic alkyne agarose beads are coupled with propargylamine to produce magnetic alkyne agarose (MAA) beads. (2) Metabolic labeling of NSPs in cultured cells is carried out with L-azidohomoalanine (AHA) and heavy- and intermediate stable isotope-labeled Lysine and Arginine. Cells with different treatment conditions are mixed and lysed by adaptive focused acoustic (AFA) sonication, using a Covaris LE220 system. (3) NSPs are enriched by covalent coupling to MAA beads via click-chemistry on a Bravo robotic liquid handling platform. NSPs are digested off the MAA beads and purified using the autoSP3 protocol. (4) NSPs are characterized by mass spectrometry via data-dependent or data-independent acquisition (DDA or DIA, respectively), and data analysis is carried out with DDA search engines such as Maxquant or DIA-NN and plexDIA for DIA data. The image of the Bravo was redrawn from the user manual using Inkscape.

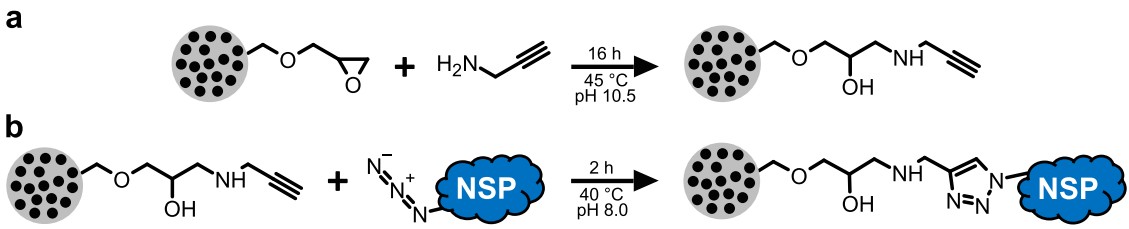

**Fig. 2 | Preparation and use of magnetic alkyne agarose (MAA) beads.**
**a** Proposed coupling mechanism of Epoxy-activated magnetic agarose beads with propargylamine to produce MAA beads. **b** AHA-containing proteins are covalently bound to MAA beads using click chemistry (Cu(I)-catalyzed azide alkyne cycloaddition (CuAAC)).

("recovery plate") for the transfer of purified peptides (Supplementary Fig. 1) for subsequent proteomic analysis.

To determine the best conditions for the use of MAA beads in the automated enrichment protocol, we generated a pulse-labeled sample and performed enrichment experiments with different amounts of MAA beads and protein input. Specifically, Hela cells were labeled with AHA and intermediate- or heavy SILAC amino acids for 4 h, and 100 µg of protein lysate was used as input for NSP enrichment with different amounts of MAA beads (2–8 µL bead volume per reaction, corresponding to 1.1–4.3% (v/v)). Although we did not compare different biological conditions at this point, we used both heavy and intermediate SILAC labels to accurately simulate the composition of such newly synthesized proteome samples. As a metric to assess the efficiency of the newly synthesized proteome enrichment, we compared the ratio of the heavy or intermediate SILAC-labeled precursor peptide ions (i.e., that originate from NSPs), over the unlabeled precursors (i.e., that originate from pre-existing proteins). For these initial optimizations, LC-MS analysis was carried out using a conventional data-dependent acquisition scheme. We observed that slightly more proteins were quantified in enriched compared to non-enriched samples (Fig. 3a, Supplementary Data 1). More importantly, the ratio of newly synthesized over pre-existing proteins was 0.25 without enrichment (i.e., −2 on log scale), while this was on average 6.5 (2.7 on log2 scale) after capture of NSPs across all tested amounts of MAA beads (Fig. 3b),

thus indicating a >25-fold enrichment of NSPs. In addition, this resulted in improved accuracy of NSP quantification in the enriched samples, determined by comparing the distribution of SILAC ratios around the expected ratio (Fig. 3c). The quantitative precision, determined by the coefficient of variation (CV) values of the SILAC ratios, was also significantly improved in enriched NSP samples (Fig. 3d).

Although the efficiency of the click chemistry-based enrichment with the presented protocol is high, we note that the use of SILAC labels is essential to distinguish genuine NSPs from non-labeled proteins that cannot be fully removed, despite stringent washing (Fig. 3a and Supplementary Fig. 2A). 'Stickyness' of unlabeled proteins does not depend on protein abundance or hydrophobicity, since both parameters span the same full range as observed for NSPs (Supplementary Fig. 2A, B), and therefore this cannot be used to exclude them from the analysis.

In addition to the comparison of different MAA bead amounts, the same sample was used to test the influence of protein input amount at a constant MAA bead volume of 4 µL (2.1% (v/v)). Across the range of tested protein input amounts (1–300 µg), we observed a consistently high ratio of SILAC labeled over unlabeled precursor intensities, indicating that the automated enrichment is efficient, even with low protein input (Fig. 3e, Supplementary Data 2). As expected, the number of identified proteins scaled with input amount, achieving >3200 proteins at 50 µg of total protein input and reaching a plateau at approximately 3600 proteins from 100 µg and upward (Fig. 3f). These

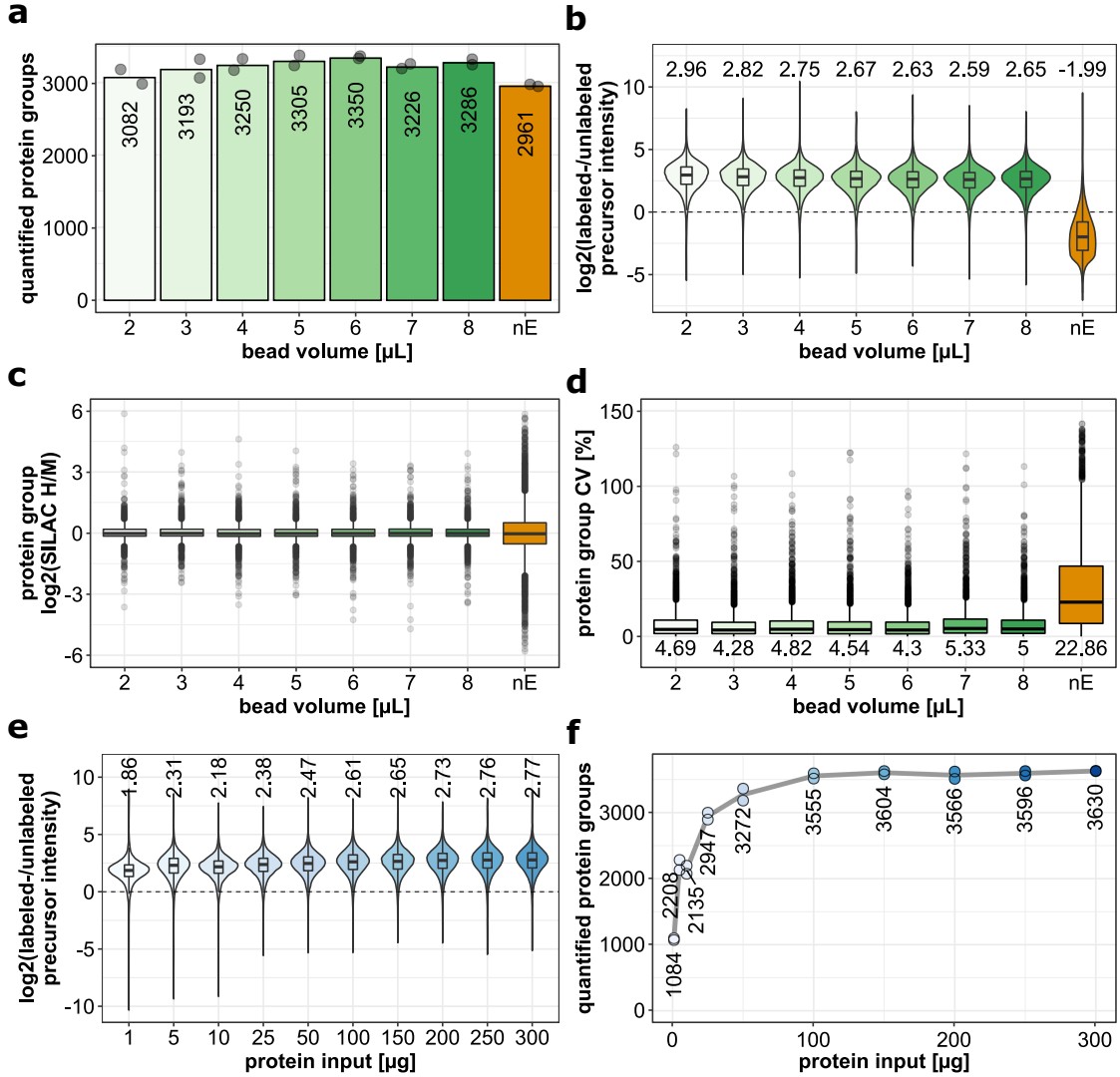

**Fig. 3 | Optimizing the semi-automated enrichment of newly synthesized proteins via QuaNPA.** Performance was evaluated when using different MAA bead volumes (panels **a**–**d**) and amounts of protein input (panels **e**, **f**). Measurements were performed using a DDA method with 105 min duration on a QExactive HF mass spectrometer (details in Supplementary Information). **a** Number of quantified protein groups (with heavy- over intermediate SILAC ratio). Numbers indicate the average of 2 replicates (gray dots) ($n = 2$, 16 samples in total). **b** Intensity ratios of heavy- and intermediate SILAC labeled precursors (originating from newly synthesized proteins), over light precursor ions (originating from pre-existing proteins). The upper and lower whiskers, of the ratio boxplots, extend from the hinges to the highest or lowest values that are within 1.5x the interquartile range. Median values are indicated in the boxplot through the horizontal lines in the center of the distributions ($n = 2$). **c** Boxplot indicating the log2 SILAC H/M ratios of the individual protein groups. The upper and lower whiskers extend from the hinges to the highest or lowest values that are within 1.5x the interquartile range. Values outside this range are plotted as dots and represent outliers. Median values are indicated in the boxplot through the horizontal lines in the center of the distributions ($n = 2$).

The expected log2 SILAC ratio is 0. **d** Coefficient of variation (CV) values of the SILAC H/M ratios of the quantified protein groups ($n = 2$). The upper and lower whiskers extend from the hinges to the highest or lowest values that are within 1.5x the interquartile range. Values outside this range are plotted as dots and represent outliers. Median values are indicated in the boxplot through the horizontal lines in the center of the distributions. **e** Intensity ratios of heavy- and intermediate SILAC labeled precursors, from NSP samples prepared with different amounts of protein input ($n = 2$). The upper and lower whiskers, of the ratio boxplots, extend from the hinges to the highest or lowest values that are within 1.5x the interquartile range. Median values are indicated in the boxplot through the horizontal lines in the center of the distributions. **f** Number of quantified protein groups across the input dilution series. Data based on 2 experimental replicates ($n = 2$, 20 samples in total). nE: non-enriched NSP sample. Color coding panels (**a**–**d**): metrics after protein enrichment of NSPs with increasing bead volumes (shades of green) or without enrichment (orange); panels (**e**, **f**): metrics after protein enrichment of NSPs with increasing protein input (shades of blue).

results indicate that 50–100 μg protein input suffices to maximize the number of protein identifications (at least for the chosen LC gradient length and MS method) and that this can be scaled down to 10 μg or less for scarce samples without a major reduction in efficiency of NSP enrichment.

Collectively, these data indicate that MAA beads permit efficient enrichment of NSPs in an automated fashion over a wide range of protein input amounts, allowing the quantification of a larger number

of NSPs with greater precision and accuracy compared to non-enriched samples. In addition, consistent performance across a range of bead quantities indicates that the approach tolerates potential variation in the amount of beads, thus contributing to experimental robustness.

**Using plexDIA for newly synthesized proteome analysis**
In addition to the described improvements of the newly synthesized proteome sample preparation via the automated enrichment with

MAA beads, we tested whether the use of data-independent acquisition (DIA) is advantageous over the use of the data-dependent analysis (DDA) used above. Increasing evidence indicates that DIA achieves increased proteome coverage in label-free approaches; however, major computational challenges and limited data analysis tools have prevented its use in combination with non-isobaric labeling methods. However, the newly developed plexDIA[44] feature of the DIA-NN software[46] was shown to maintain quantitative accuracy for the analysis of mTRAQ-labeled samples, while archiving depth comparable to label-free DIA[44]. We therefore aimed to test the performance of DIA-based MS and plexDIA for the analysis of SILAC-labeled samples, and in particular for NSPs. To this end, we prepared two benchmark samples with defined ratios of SILAC-labeled Hela cell lysates for comparative analysis via a conventional DDA method and two DIA methods. The first sample (mix1) consisted primarily of light protein (70%), with only a smaller fraction of intermediate- and heavy-labeled proteins, mimicking a conventional pulse-labeled sample without enrichment. The second sample (mix2) consisted of a smaller fraction of light proteins (20%), with a major contribution of labeled protein, thus mimicking an enriched newly synthesized proteome sample (Fig. 4A). Since, in contrast to DDA, both MS1- and MS2-based quantitative data can be used for the analysis of the DIA measurements, two DIA methods were designed: DIA method 1 (DIA m1) was optimized for short cycle time and MS2-based quantification, whereas DIA method 2 (DIA m2) was optimized for MS1-based quantification by including additional high-resolution MS1 scans, resulting in a slightly increased cycle time (Fig. 4B, see "Methods" for details). The same LC method was used in all cases. To conduct a thorough comparison of the different acquisition methods, samples, and analysis software, we determined the number of protein identifications, quantitative precision and accuracy for each of the analyses.

The DDA approach consistently quantified fewer proteins than either of the DIA methods in both samples (Fig. 4C, Supplementary Data 3). In addition, this number was constant among all L/M/H ratios, indicating that the ability to quantify is independent of the abundance of each of these channels (within the tested range). More proteins were quantified in the samples of mix 1, which was primarily composed of unlabeled light protein. A number of observations can be made in the DIA data: first, more proteins were quantified in mix2 than in mix1, possibly explained by the difficulty that plexDIA has to quantify the more extreme ratios in mix 1. Second, the number of quantified proteins varied depending on the SILAC channels. Notably, fewer proteins were quantified in cases where the lower abundance channel was involved (H and M in mix 1, L in mix2). These observations suggest that those proteins are credited that occur in higher abundance and at a ratio closer to 1. This is as expected since it is more difficult to quantify signals at a low S/N ratio, and the gained proteins in the DIA data compared to DDA are on average of lower intensity (Supplementary Fig. 4). Third, quantification by MS1 (blue) and MS2 (red) produced very similar numbers of quantified proteins, with a slight tendency for more identifications by MS2 quantification, which was observed both in DIA m1 and m2. Finally, one of the most striking observations from the data is that the number of H/M-quantified proteins in mix2 more than doubled from approximately 3000 in DDA to well over 6000 in DIA (Fig. 4C), indicating a favorable scenario for newly synthesized proteome samples that are dominated by these two SILAC labels (also see Fig. 3b).

In addition to the number of quantified proteins, the precision of the quantified protein groups was compared using CV values (Fig. 4D). All median CVs were in a narrow range between 5 and 15%, indicating excellent precision across all data acquisition methods. Yet some subtle trends were observed, where quantification by DDA was more precise in mix2 than in mix1, possibly because ratios were less extreme. Quantitative precision in the two DIA methods at the MS1 level (blue) was highly comparable both for mix1 and mix2 (11–15%), with a

tendency for improved precision by DIA m2, likely benefiting from the additional MS1 scans that were included in the method for this purpose. Clearly, MS2-based quantification (red) yields data with even greater precision, and the lowest CV (5.4%) was obtained for H/M ratio with the MS2-optimized DIA method (Fig. 4D). Next, to assess the accuracy of protein quantification, SILAC ratios were plotted and compared to the expected ratios of the benchmark samples (Fig. 4E). The accuracy of the MS1-based quantification data (blue) from both DIA methods is comparable to the DDA data but also includes larger numbers of outlier values. As observed for precision above, the accuracy is greater for less extreme SILAC ratios, and the best accuracy was achieved for MS2-based quantification of 1:1 ratios (H/M) both in mix1 and mix2, which actually produced the most accurate data across the entire dataset (Fig. 4E). These results are similarly reflected on the precursor level (Supplementary Fig. 5).

Overall, this benchmark dataset indicates that DIA measurements of SILAC labeled samples, in combination with the plexDIA functionalities of DIA-NN, yield high-quality data with drastically increased proteomic depth and robust quantification. This approach benefits from the large fraction of labeled peptides in the samples, making it highly suitable for the analysis of enriched NSP samples.

Through a combination of the semi-automated NSP enrichment and plexDIA, increased proteomic depth can be archived even with relatively low input amounts for enriched NSP samples. Similarly to the previous measurements in DDA mode, high intensity ratios of labeled precursors are detected, indicating efficient click-enrichment and quantification of NSP, even at low overall protein input. Increased input leads to an increase in protein and peptide identifications, but a maximum value of around 6000 quantified protein groups is archived ≥100 μg. However, even with 25 μg input >5000 protein groups can be quantified (with a 90 min DIA method on a Q Exactive HF) (Supplementary Fig. 6).

## Using QuaNPA for the analysis of newly synthesized proteome changes in response to IFNg

Having established QuaNPA as an optimized workflow for newly synthesized proteome analysis that includes the generation of magnetic alkyne beads, conditions for automated protein capture and clean-up, DIA-based mass spectrometry, and data analysis by plexDIA, we aimed to demonstrate its utility to understand proteome response to cellular perturbations. Specifically, we studied the response of Hela cells to interferon-gamma (IFNg) in a time-resolved manner to identify proteins that are induced by this immune-stimulatory factor. Therefore, we treated Hela cells with IFNg and collected cells at five different time points (2–24 h), each with a 0.5% (w/v) BSA-treated control, and all in 3 replicates (i.e., totaling 30 samples). Notably, cells for each sample were grown in one well in a 6-well plate which sufficed to obtain >50 μg total protein per condition. Importantly, cells were lysed and proteins extracted by adaptive focused acoustic (AFA)-ultra-sonication for all samples simultaneously in a 96-well plate[45], followed by automated NSP enrichment and peptide clean-up with automated SP3 both on a Bravo liquid handling platform. Thus, this collectively constitutes an integrated multistep workflow to process cells to purified peptides, with minimal manual intervention (Fig. 5a). Upon NSP analysis by DIA mass spectrometry (90 min method, 70 min active gradient) and data analysis by plexDIA, >6000 proteins were quantified per time point and replicate, overall totaling 8130 protein groups (6887 unique proteins) (Fig. 5b, Supplementary Data 4). The rate of missing values remained at a moderate 18.11% across the whole dataset, and on average was 9.58% for individual samples, reflecting a key merit of DIA. The precision of quantification (6–8%, Fig. 5c) was within the same range as for the benchmark dataset (Fig. 4D). Shorter labeling times did not lead to a noticeable reduction in the number of quantified proteins or precision (Fig. 5b, c), indicating that even sparsely labeled proteins were confidently identified and quantified. Using principal

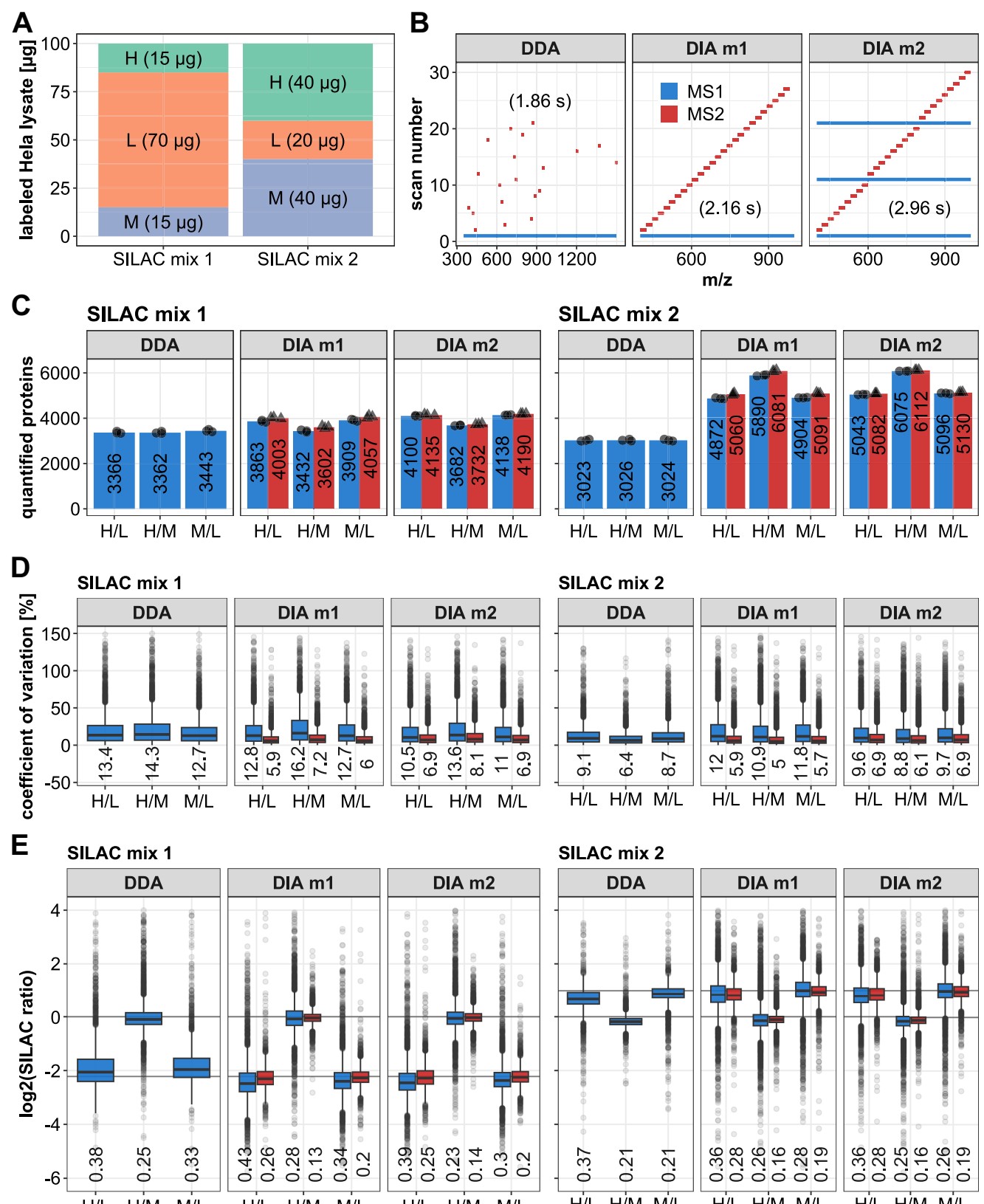

component analysis (PCA), the individual replicates of the samples clustered together for each time point, following the time course of IFNg treatment, indicating a progressive and reproducible effect in the newly synthesized proteome (Fig. 5d). This was confirmed in a differential protein expression analysis, where multiple significant changes in the newly synthesized proteome were detected at each time point,

even as early as 2 h (Fig. 5e) and Supplementary Data 5. Most of the significantly upregulated proteins are known downstream targets of IFNg or are directly involved in the IFNg signaling pathway (Fig. 5f), including ICAM1, STAT1 and TAP1 which were among the earliest detected proteins, already observed after 2 h and 4 h (Fig. 5e). Compared to published newly synthesized proteome analysis data of

**Fig. 4 | Comparative analysis of SILAC labeled benchmark samples using data-dependent acquisition (DDA) and data-independent acquisition (DIA) mass spectrometry. A** Schematic representation of the composition of the SILAC-labeled Hela samples. **B** Schematic representation of the top 20 DDA method and the two DIA methods, which were used for the analysis. The maximum cycle time for each method is indicated in brackets. **C** Comparison of the number of quantified protein groups for the different methods, with values based on MS1-based quantification in blue and MS2-based quantification indicated in red. Numbers indicate the average of 3 replicates (indicated individually with dots and triangles ($n = 3$)). **D** Boxplot indicating coefficient of variation (CV) values of the SILAC ratios of the quantified protein groups, with values based on MS1-based quantification in blue and MS2-based quantification indicated in red ($n = 3$). Median values are indicated

below the boxplots. The upper and lower whiskers extend from the hinges to the highest or lowest values that are within 1.5x the interquartile range. Values outside this range are plotted as dots and represent outliers. **E** Boxplots indicating the distribution of log2-transformed SILAC ratios of the quantified protein groups, with values based on MS1-based quantification in blue and MS2-based quantification indicated in red ($n = 3$). The median of the difference from the theoretical log2 ratio is indicated below the boxplots. Upper and lower whiskers extend from the hinges to the highest or lowest values that are within 1.5x the interquartile range. Values outside this range are plotted as dots and represent outliers. Data are based on 3 technical replicates of the single SILAC mix samples (2 different samples measured with 3 different LC-MS methods in 3 technical replicates ($n = 3$, 18 measurements in total)).

IFNg-treated Hela cells, prepared with the PhosID methodology[24], the QuaNPA workflow achieves greater proteome coverage and higher sensitivity in the detection of characteristic IFNg-induced protein expression changes (Supplementary Fig. 7). Specifically, we identified 2.5-fold more proteins from 5-fold lower protein input and were able to identify various IFN-related gene sets at the 4 h time point that were missed by PhosID at this early time point (Supplementary Fig. 7), indicating the sensitivity and efficiency of the QuaNPA workflow. However, fewer differentially regulated proteins were detected at the 4 h IFNg treatment time point, compared to data produced with the PhosID workflow. Furthermore, when comparing our data to previous work studying IFNg response at the transcriptome level[47], across all time points we observed a significant enrichment of proteins whose mRNA was upregulated (Supplementary Data 6), and whose gene promoters were bound by STAT1 (Fig. 5f, Supplementary Data 7). Indeed, a modest positive correlation was observed between mRNA and NSP expression ($R = 0.43$), which was increased when only considering proteins that were differentially expressed ($R = 0.66$) or when only considering STAT1 targets ($R = 0.54$). Notably, several differentially expressed transcripts did not lead to a change in protein expression, and conversely, multiple differentially synthesized proteins had no significant change in their mRNA abundance (Supplementary Fig. 8), indicating the role of translational regulation.

By a combination of the data from different time points of IFNg treatment, we evaluated temporal profiles of differentially expressed NSPs. Proteins were assigned to 3 groups to classify their early (2 h), intermediate (4–9 h) or late response to IFNg (24 h), depending on the earliest time point that differential expression was observed with statistical significance (absolute log2 fold change > 1 and adj. $p$-value < 0.05), at any of the 5 time points and CV < 20% across all time points; Supplementary Data 8. Furthermore, for these proteins we collected additional evidence for IFNg regulation, by listing the number of reported datasets in the interferome.org database[48] that provided evidence for their regulation by IFNg, in human cells (Fig. 6a). At the early time point of 2 h IFNg treatment, a small set of proteins is differentially expressed, including ICAM1, SOD2 and STAT1, which is the major transcriptional mediator in the IFNg signaling pathway. These three proteins are well-established targets of IFNg signaling with numerous reports of induced expression upon IFNg stimulation, establishing them as known IFNg hallmark proteins. Additionally, strong and rapid upregulation was observed for ZC3HAV1 (PARP13), a protein with a strong anti-viral function and prominently reported in the interferome as a target of IFNg, although it is not listed as an IFNg-response gene according to GSEA-Hallmark or gene ontology data (Fig. 6a). Interestingly, differential expression of the adapter protein TICAM1 was only detected at the 2 h and has only been mentioned in very few reports in relation to IFNg, making it a potential novel or at least under-studied candidate as an IFNg-responsive protein. Proteins with differential expression 4–9 h after IFNg stimulation also included multiple well-characterized targets of IFNg, such as endogenous peptide antigen transporter TAP1, its interaction partner TAPBP and TAP2, MHC class-I component HLA-E, tryptophan tRNA-ligase WARS1,

transcription factor SP110 and phospholipid scramblase PLSCR1. The group of proteins that showed a delayed response to IFNg (24 h) primarily consists of proteins associated with functions of the immune system. Complement proteins C1S and C3 and proteins involved in endogenous antigen presentation via MHC class I, including HLA-A, HLA-C, HLA-H, and PSME1. In addition, other hallmark proteins of IFNg response were found in this category, such as IFIH1, IFI30, PML, and PARP12, reaching maximum levels late (24 h) after IFNg treatment, although for nearly all of them, expression gradually increased for the duration of IFNg exposure (Fig. 6a). Beyond this, we identified differential expression of several more proteins which have not been previously reported as IFNg targets, such as KLF3 and SIN3B (Fig. 6a, Supplementary Data 5). Interestingly, both these transcription factors have established roles in hematopoiesis[49,50], with prior evidence of being regulated by IFNg[51,52], despite not being listed in interferome.org. These data show that QuaNPA identifies bona fide targets of IFNg, in line with IFNg's known role in inducing hematopoiesis.

Apart from proteins whose expression is induced by IFNg, we observed a smaller group that is repressed, such as CXCR4 (Fusin) and IER3 (Fig. 6a). IFNg-induced downregulation of CXCR4 was previously shown to result in reduced tumor metastasis and virus replication[53]. In addition, we observed reduced synthesis of several proteins that to the best of our knowledge have not been reported as IFNg responsive proteins, including stark and immediate (2 h) downregulation of the mitochondrial protein MT-ATP8, and a more gradual decrease of BICD2, S100A6 and SUMO1. Interestingly, these latter three proteins have previously been shown to negatively regulate STAT1 signaling: depletion of BICD2 was shown to increase levels of STAT1 mRNA[54], S100A6-knockdown lead to increased protein levels and phosphorylation of STAT1[55], and SUMO1-conjugation of STAT1 lead to reduced levels of STAT1 phosphorylation and transcription of IFNg response genes[56]. Therefore, our data indicates that IFNg-induced downregulation of these targets constitutes a feed-forward mechanism to enhance IFNg signaling output.

To validate the potential new IFNg targets, we performed a targeted proteomic analysis using label-free parallel reaction monitoring (PRM). Relative changes in target protein abundance were analyzed after 4 h and 24 h IFNg or control treatment. Canonical IFNg target proteins STAT1, TAP1 and ICAM1 were included as positive controls for the PRM analysis. Interestingly, only the latter was also among the initially upregulated NSP after 2 h treatment with IFNg (Fig. 5e) and was borderline significant at the 4 h time point, highlighting the sensitivity of QuaNPA (Fig. 6b). In addition to the strong upregulation of TAP1, STAT1 and ICAM1 after 24 h treatment with IFNg, moderate upregulation of SIN3B and downregulation of BICD2 protein abundance could be measured via PRM, matching the results of the newly synthesized proteome analysis. However, no significant changes in protein abundance could be measured for the remaining candidate IFNg targets (SUMO1, S100A6 and CRK) (Fig. 6a, b and Supplementary Fig. 9).

Collectively, the QuaNPA workflow allowed us to quantify proteome-wide changes in protein synthesis in response to IFNg in a

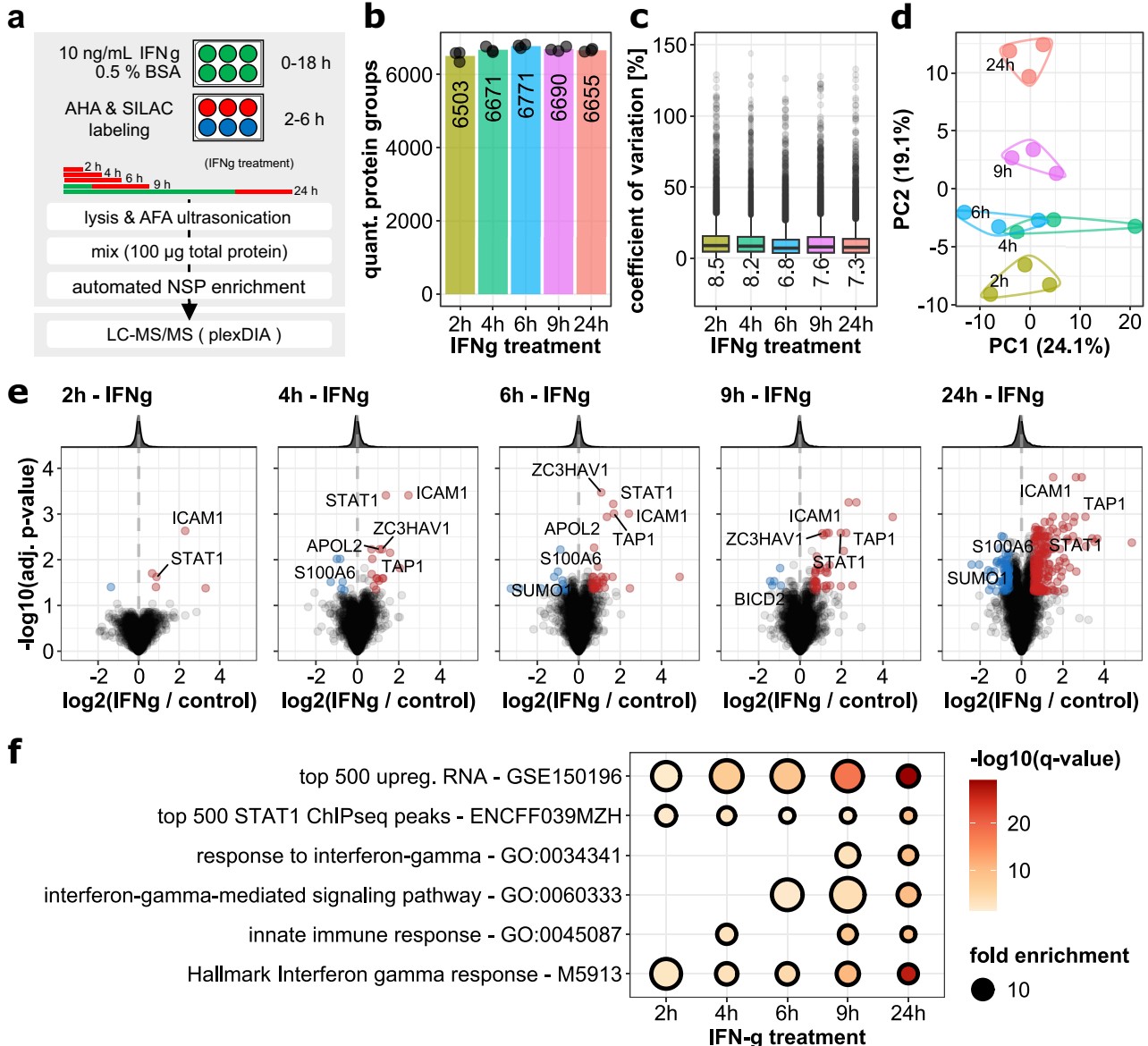

**Fig. 5 | Analysis of newly synthesized proteins in Hela cells in response to treatment with interferon-gamma (IFNg). a** Schematic representation of the experimental design and analysis workflow. The IFNg treatment time is indicated using text labels and the length of the colored bars, the pulsed SILAC and AHA metabolic labeling time is indicated with red coloration of the bars. **b** Number of quantified protein groups in the samples with the indicated IFNg treatment time points. Numbers indicate the average of three replicates (*n* = 3) (indicated individually with black dots). **c** Boxplots indicating the coefficient of Variation (CV) values of the quantified protein groups for each time point (*n* = 3). Upper and lower whiskers extend from the hinges to the highest or lowest values that are within 1.5x the interquartile range. Median values are indicated in the boxplot through the horizontal lines in the center of the distributions. Values outside this range are plotted as dots and represent outliers. **d** Principal component analysis (PCA) of the NSP samples (*n* = 3). Colors in panels (**b**–**d**) indicate treatment time points.

**e** Volcano plots of the IFNg-induced changes in the newly synthesized proteome. Significantly upregulated proteins (adjusted *p*-value < 0.05 and log2 fold change > 0.585) are highlighted in red and significantly downregulated protein (adjusted *p*-value < 0.05 and log2 fold change < −0.585) are highlighted in blue. Only unique protein groups were included in the differential expression analysis. A modified empirical Bayes moderated *t*-test, adjusting t-statistic and two-sided *p*-values with precursor counts, was carried out with the "spectraCounteBayes" function of the DEqMS R/Bioconductor package[69]. Data based on 3 experimental replicates (*n* = 3, 15 samples in total). **f** Dot plot highlighting overrepresented sets of proteins, which are significantly upregulated in response to IFNg at the indicated time points (*q*-value < 0.05). Overrepresentation analysis was carried out using a hypergeometric test via the "enricher" function of the clusterProfiler R/Bioconductor package[70]. Adjustments for multiple comparisons were carried out using the Benjamini–Hochberg approach[73].

time-resolved manner, benefitting from automated sample preparation and reduced input requirements. We detected changes in the newly synthesized proteome at various time points during the time course of IFNg treatment, distinguishing immediate and delayed events occurring at 2 h or gradually toward 24 h, indicating primary and secondary targets of IFNg signaling. Importantly, this included induction or repression of established IFNg target proteins as well as several novel ones, two of which could be validated in targeted

proteomic analysis, demonstrating the power of QuaNPA to infer relevant biology from temporal expression profiling of newly synthesized proteins.

## Discussion

The analysis of newly synthesized proteins affords distinct conceptual advantages over conventional protein abundance profiling, giving insight into the cell's proteome response to perturbations at rapid time

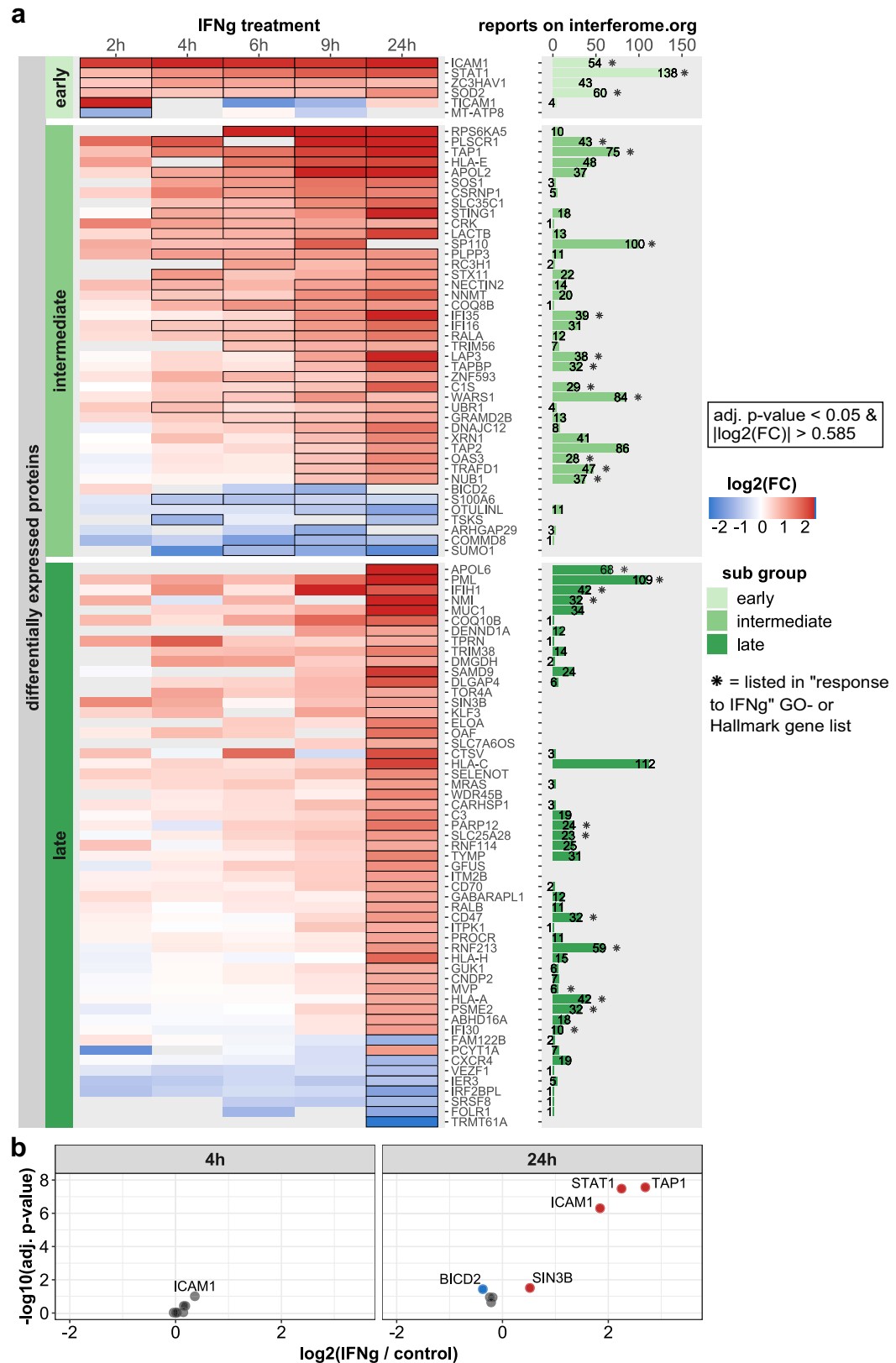

scales. Yet, involved protocols for labeling of cells, enrichment of NSPs, and analysis of data result in a labor-intensive methodology with low throughput and sensitivity, thus hampering adoption in the field. In QuaNPA, we have addressed these issues while maintaining the benefits of AHA labeling, protein enrichment, and pulsed SILAC labeling. We have achieved this by designing high-capacity alkyne-

functionalized magnetic sepharose beads, automating protein enrichment, implementing a single-shot DIA-based LCMS methodology, and using DIA-NN's plexDIA-function for data analysis. Moreover, we integrated this with the autoSP3 protocol that we developed recently[45], combining multiplexed cell lysis by AFA-based sonication and protein/peptide clean-up by SP3 on the same Bravo liquid

**Fig. 6 | Response to IFNg treatment. a** Heatmap of newly synthesized proteins with differential expression in response to IFNg treatment. The subset of differentially expressed proteins (|log2 fold change| > 1 and adj. *p*-value < 0.05, at any of the 5 time points and CV < 20% across all time points) was selected and classified into three groups depending on the earliest time point in the time course their change in expression reached statistical significance (|log2 fold change| > 0.585 and adj. *p*-value < 0.05) (early: 2 h; intermediate: 4–9 h; late: 24 h). Significant changes in newly synthesized protein abundance are indicated with black outline in the cells of the heatmap and the color gradient indicates the log2 fold change values of the proteins. The bar graph indicates the number of datasets in the interferome database, in which a fold change >2 or <0.5 was reported in human cells or tissues for the respective protein or gene. The asterisk sign (∗) on top of bars indicates whether the protein is listed in the response to IFNg Hallmark or GO-term. Statistical

analysis of the quantitative NSP data was carried out using a modified empirical Bayes moderated *t*-test, adjusting t-statistic and two-sided *p*-values with precursor counts, via the "spectraCounteBayes" function of the DEqMS R/Bioconductor package[69]. **b** Volcano plots of the IFNg-induced protein abundance changes, after 4 h and 24 h treatment, determined via targeted PRM analysis. Significantly upregulated proteins (adjusted *p*-value < 0.05 and log2 fold change > 0.585) are highlighted in red and significantly downregulated protein (adjusted *p*-value < 0.05 and log2 fold change < −0.585) are highlighted in blue. Statistical analysis was carried out using TIC-normalized fragment peak areas, via the MSstats module in Skyline. Data are based on 3 experimental replicates (*n* = 3, 12 samples in total). Statistical analysis in MSstats is carried out using by fitting a two-sided mixed linear model onto the data, via the "lm" and "lmer" functions of the stats and lme4 R packages.

handling platform used for NSP enrichment. Collectively, QuaNPA thus constitutes an integrated pipeline with minimal manual intervention, and processing of samples in the same plate throughout. As a result, the semi-automated enrichment of 96 samples is completed in 5.5 h for 96 samples (4.25 h for 8 samples), including the 2h-incubation of the CuAAC-based coupling of the NSPs to the magnetic alkyne beads, and subsequent washing steps that make up most of the protocol runtime. Tryptic digestion of the samples was carried out off the Bravo deck since it requires a heated lid. Lyophilization of the diluted tryptic peptides can be carried out in approximately 1.5 h, and subsequent peptide purification via autoSP3 takes about 1.25 h of runtime, all for 96 samples simultaneously. QuaNPA requires low amounts of input material (≤100 µg) to obtain high-quality data (Fig. 3e, f, Supplementary Figs. 2 and 6), which enables the use of small culturing flasks/ dishes such as 6-well plates, thus rendering large-scale studies with multiple cellular treatments a manageable endeavor.

When evaluating QuaNPA using a pulsed-SILAC and AHA-labeled HeLa sample, we found that protein enrichment on MAA beads modestly increased the total number of identified NSPs compared to the omission of enrichment (Fig. 3a), which is in line with our previous observations[5]. More importantly, the enrichment of NSPs improved the quantification of newly synthesized proteome analysis (Fig. 3c, d). Both observations can be readily explained by the removal of the excess of pre-existing proteins, increasing the signal-to-noise of peptides originating from AHA-containing proteins. Through the usage of the MAA beads in a semi-automated protocol, the efficiency of NSP enrichment was maintained across a broad range of protein input amounts (1–300 µg, Fig. 3e), enabling diverse study designs both with low or non-limited cell numbers. As observed in this study, even stringent washing steps during NSP enrichment cannot completely remove unlabeled pre-existing proteins. These can be readily recognized by their SILAC labeling status, contrasting with label-free approaches[15,24,57,58] where such contaminants cannot be distinguished from genuine NSPs, thereby compromising quantitative accuracy.

The recently developed plexDIA functions of DIA-NN enable the DIA-based analysis of multiplexed samples with non-isobaric labels[44]. Since the analysis of label-free samples via DIA has been demonstrated to achieve deep proteome coverage in single LC-MS runs (i.e., without peptide fractionation), similar performance for SILAC-labeled proteins would save measurement time, and increase the throughput of the QuaNPA workflow. Indeed, we observed a major increase in the number of identified proteins in our benchmark sample when using either of the tested DIA methods, doubling the number to 6000 proteins compared to DDA analysis (Fig. 4C). Accordingly, very similar numbers were achieved when analyzing the actual NSP samples (Fig. 5b and Supplementary Fig. 6). These results were obtained without optimization of other parameters in DIA-NN. In particular, and in contrast to conventional DDA SILAC search engines, DIA-NN does not feature a "re-quantify" function, which enables the calculation of SILAC ratios with labeled peptides whose intensity is close to the noise level. Instead, DIA-NN calculates channel and peak translation *q*-values,

which at high stringency lead to the filtering of labeled peptide ions with very low intensity. Our benchmark showed that plexDIA performed particularly well for triple-SILAC samples that resemble enriched NSPs. In addition, we observed high quantitative accuracy and precision when using plexDIA both with MS1- and MS2-optimized quantification, showing comparable performance to conventional DDA analysis. (Fig. 4D, E). Since we obtained high-quality data with stringent translated- and channel *q*-value filters, we did not extensively investigate the different features of the DIA-NN output tables such as the multiple quantitative metrics and *q*-value filters. Optimization of these parameters might further improve the proteomic depth and quantitative accuracy of samples with large SILAC ratios, as in our mix1. To address the issue of large SILAC ratios, the inverted spike-in workflow, implemented in the Spectronaut software was used for pSILAC DIA analysis[43,59]. In recent work using different instruments for plexDIA, such as the timsTOF SCP, MS1-based quantification was preferred over MS2-based quantification[44]. This somewhat contrasts with our observations (Fig. 4); however, this may result from the use of a different mass spectrometer, the use of low-input samples at the single-cell level, and the fact that samples were labeled using different methods. Future work should further delineate the best approach for diverse use cases, but nevertheless, these published data, now augmented with our SILAC data, demonstrate that non-isobaric labeling can be effectively used in DIA-based workflows for multiplexed proteomic experiments. The capacity to analyze NSP samples in single LC-MS runs has important implications since it enhances throughput by significantly reducing measurement times. These characteristics make the QuaNPA workflow ideally placed for quantitative NSP analyses across multiple conditions, such as different cellular treatments or time course experiments.

We applied QuaNPA to investigate the cellular response of HeLa cells to IFNg since this is a well-characterized perturbation with many known targets to allow benchmarking of our data. For instance, the biological system has also been used in global proteome and transcriptome analysis, and in two recent strategies coupling AHA-containing NSPs to phosphonate alkynes[24] or biotin-alkyne[59] with subsequent affinity purification via a Bravo AssayMap platform. However, both protocols require relatively large amounts of protein input (500 µg per sample/condition[24], i.e., 10x more than in QuaNPA), or lysates from cells grown in 15 cm dishes[59], and they contain long additional steps, such as overnight dephosphorylation[24] or dialysis[59], likely leading to protein losses and contributing to the reduced proteomic depth and sensitivity compared to QuaNPA (Supplementary Fig. 7). In our data we observed the upregulation of multiple well-established downstream targets of IFNg signaling. Importantly, core response proteins (ICAM1, STAT1, SOD2) were already observed after 2 h of IFNg treatment, i.e., much earlier than in previous work[24,59]. In addition, many other IFNg-targets exhibited a gradual increase in protein synthesis over time, including proteins for which we could not find prior association with IFNg signaling[48]. We anticipate that proteins observed to be immediately regulated (2–4 h) may be direct targets of

IFNg, including those that go down in expression such as SUMO1, S100A6 and MT-ATP8 (Fig. 6a). Others may also represent secondary responses, regulated by the large number of IFNg hallmark proteins induced during the time course. Moderate changes in BICD2 and SIN3B abundance could be validated using a targeted proteomics approach. The lack of significant protein abundance changes for the other tested targets (SUMO1, S100A6 and CRK) could be due to the fact that more time is required for the changes in protein synthesis to lead to a measurable effect in overall protein abundance. (Fig. 6a, b and Supplementary Fig. 9). Alternative mechanisms of protein degradation or stabilization could also explain the discrepancy in the results. However, this would need to be further investigated.

The QuaNPA workflow was designed to determine changes in protein synthesis as a measure for a perturbation-induced proteome response; however, it does not take into account the role of protein degradation that may act as a counteracting effect to reach a certain overall protein expression level. Pulse-chase experiments can be performed to obtain a more complete view that considers both synthesis and degradation, usually in a time-course manner, and enrichment of labeled proteins via the QuaNPA workflow could readily be adapted for such applications. Furthermore, by preparation of magnetic azide agarose beads, proteins labeled with other non-canonical amino acids such as L-homopropargylglycine (HPG), β-ethynylserine (β-ES)[60] or puromycin conjugate O-propargylpuromycin (OPP) could be used instead of AHA, expanding the utility of the QuaNPA workflow (Supplementary Fig. 10).

In conclusion, the QuaNPA workflow features automated sample preparation of up to 96 samples in parallel, enabling the detection of changes in protein synthesis with high quantitative accuracy and precision with increased proteome coverage, while requiring short metabolic labeling and LC-MS/MS measurement times. Although we used AHA here as a mark for protein NSP enrichment, it is readily conceivable to implement other clickable amino acids or puromycin analogs in the workflow. In addition, performance characteristics of QuaNPA should enable combined studies with global protein expression profiling, to benefit from complementary insights that can be gained from alterations in protein synthesis and in overall protein abundance. We anticipate that QuaNPA will empower large-scale NSP analyses in numerous biological contexts to understand the proteomic response that is elicited by diverse perturbations and signaling events.

## Methods
### Preparation of magnetic alkyne agarose beads
Epoxy-activated magnetic agarose beads (Cube Biotech) were coupled with propargylamine (Santa Cruz Biotechnology) to produce magnetic alkyne agarose (MAA) beads. Then, 5 mL epoxy-activated magnetic agarose beads were washed with 10 mL milliQ water and resuspended in the coupling solution of 1 M propargylamine in 0.5 M dipotassium phosphate solution (pH 10.5). The handling of the propargylamine and coupling solution was carried out under a fume hood with appropriate safety precautions. The beads and coupling solution were incubated in a thermo shaker at 45 °C for 16 h, and beads were then washed with 25 mL milliQ water. To ensure the complete quenching of the remaining epoxy groups on the beads, they were incubated with 1 M Tris-HCl buffer (pH 8.0) for 4 h. The beads were subsequently washed with 45 mL milliQ water and stored in 20 mM sodium acetate buffer (pH 6.5) with 20% ethanol at 4 °C. The MAA beads are stable at 4 °C for multiple months.

### Cell culture
Hela cells, obtained from ATCC (CCL-2), were grown in DMEM high glucose medium (Gibco) supplemented with 2 mM L-glutamine, 10% (v/v) fetal bovine serum (Gibco) and an additional 2 mM GlutaMAX (Gibco). For the interferon-gamma (IFNg) stimulation experiments,

Hela cells were treated with 10 ng/mL recombinant IFNg (Cell Signaling), diluted in 0.5% (w/v) bovine serum albumin (Serva). For the preparation of the SILAC benchmark samples, Hela cells were grown in high glucose DMEM, with the previously listed supplements, dialyzed fetal bovine serum (Gibco) and heavy- ($^{13}C_6$$^{15}N_4$-Arg, $^{13}C_6$$^{15}N_2$-Lys), intermediate ($^{13}C_6$-Arg, $D_4$-Lys) or light isotope-containing Lysine, Arginine for 10 days. Hela cells were grown in 15-cm dishes for the preparation of newly synthesized proteome samples used for protocol optimization. For the SILAC benchmark sample preparation, the cells were grown in 10-cm dishes and for the preparation of the samples of IFNg-treated Hela cells, the cells were grown in 6-well plates. No cell line authentication and testing for Mycoplasma contaminations was carried out.

### Metabolic labeling of newly synthesized proteins
A metabolic labeling approach, combining pulsed stable isotope-labeling (pSILAC) and L-Azidohomoalanine (AHA)-based labeling of newly synthesized proteins was used. Prior to the labeling, the cells were washed with warm PBS and incubated with DMEM high glucose medium deprived of Methionine, Arginine and Lysine for 45 min. The pulsed SILAC and AHA labeling was carried out with Methionine-free DMEM high glucose medium containing heavy- ($^{13}C_6$$^{15}N_4$-Arg, $^{13}C_6$$^{15}N_2$-Lys) or intermediate ($^{13}C_6$-Arg, $D_4$-Lys) Lysine, Arginine and 100 µM AHA for 2 h, 4 h or for a maximum of 6 h. The cells used for initial method optimization were labeled for 4 h without any perturbations. The IFNg-stimulated cells were labeled for the indicated IFNg treatment time, except for the 9 h and 24 h time points, in which the cells were labeled during the last 6 h of IFNg exposure.

### Cell lysis and ultrasound sonication
Cells were lysed with lysis buffer containing 1% Sodium-dodecylsulfate (SDS), 300 mM HEPES (pH 8.0) and cOmplete EDTA-free protease inhibitor cocktail (Merck). Lysates, which were used for the method optimization and SILAC benchmark were sonicated with a probe sonicator (Branson) at 10% power for 1 min. The IFNg-treated cells were sonicated in AFA-tube TPX strips (Covaris), using a Covaris LE220R-Plus for 300 s at 325 peak power with a duty factor of 50%, 200 cycles per burst, average power of 162.5. The dithering parameters were set to ± 5 mm in $x$ and $z$ direction and 4.5 mm in $y$ direction at a speed of 20 mm/s. Protein concentrations of the sonicated lysates were determined using a BCA assay (Pierce). A total of 100 µg (50 µg per condition) protein was used as input for the enrichments, except in dilution series to test performance at lower input.

### Automated enrichment of newly synthesized proteins
The automated newly synthesized proteome enrichment protocol was programmed to enable the processing of up to 96 samples in parallel in a PCR plate. By setting the number of columns on the sample plate, incubation times, volumes of buffers and reagents as variables, adaptations to the protocol can easily be introduced.

The combined lysates were diluted to a total volume of 150 µL using lysis buffer. In order to prevent the coupling of proteins containing strongly nucleophilic Cysteine to the beads[61], the samples were alkylated by the addition of 3.4 µL of 600 mM iodoacetamide (IAA) for 20 min at room temperature. Subsequently, 20 µL of magnetic alkyne agarose (MAA) beads, diluted in lysis buffer and the Copper(I)-catalyzed Azide Alkyne Cycloaddition (CuAAC) reaction[62,63] mixture were added, containing 21.62 mM CuSO₄, 108.11 mM Tris-hydroxypropyltriazolylmethylamine (THPTA), 216.22 mM pimagedine hydrochloride and 216.22 mM sodium ascorbate. Next, the plate was removed from the Bravo platform, sealed using VersiCap Mat 96-well flat cap strips (Thermo Fischer Scientific) and incubated for 2 h at 40 °C in a thermal shaker. Following the coupling of the AHA-containing newly synthesized proteins (NSP), the plate was unsealed

and moved back onto the orbital shaker (position 9, Supplementary Fig. 1) on the Bravo platform, and the supernatant was removed, by placing the sample plate on a magnetic rack (ALPAQUA MAGNUM FLX enhanced universal magnet) for 30 s and aspirating the supernatant in two steps using tips from position 6 and dispensed in the waste plate (position 2). The beads were subsequently washed with 150 μL milliQ water. After the addition of the liquid to the beads on the orbital shaker, the plate was moved to the heating station (position 4) where the beads were mixed by pipetting up and down 8 times with a constant flow rate of 300 μL/s, to prevent aggregation of the beads. Next, the plate was transferred onto the magnetic rack, and the supernatant was removed. The NSP bound to the beads were subsequently reduced and alkylated by the addition of 150 μL of 10 mM Tris(2-carboxylethyl) phosphine (TCEP) and 40 mM 2-chloroacetaminde (CAA), dissolved in 100 mM Tris-HCl buffer (pH 8.0), containing 200 mM NaCl, 0.8 mM Ethylendiamintetraacetic acid (EDTA), 0.8% SDS and incubating on the heating station at 70 °C for 20 min and subsequent incubation at 20 °C for 15 min on the orbital shaker. The beads were subsequently washed three times with 1% SDS dissolved in 100 mM Tris-HCl (pH 8.0), 250 mM NaCl and 1 mM EDTA buffer, once with milliQ H$_2$O, three times with 6 M Guanidine-HCl in 100 mM Tris-HCl (pH 8.0) and three times with 70% ethanol, in consecutive washing steps of 150 μL each. Following the washing steps, the beads were resuspended in 50 μL 100 mM Ammonium bicarbonate buffer (pH 8.0). Proteins were digested off the beads by adding 6 μL of 1 μg/μL sequencing grade Trypsin (Promega), diluted in 50 mM acetic acid, for 16 h at 37 °C, which was performed in a thermal shaker after sealing the plate with VersiCap Mat 96-well flat cap strips.

## Automated peptide purification

The automated SP3 protocol was used for the purification of peptides by processing 16–96 samples in parallel on a Bravo liquid handling robot[45]. Following protein digestion, the peptide-containing supernatant was transferred from the sample plate on position 7 onto a new plate on position 8, and peptides were lyophilized using a UNIVAPO-150H vacuum concentrator, coupled to a UNICRYO MC2 cooling trap and UNITHERM 4/14 D closed circuit cooler (UNIEQUIP). On the Bravo platform, magnetic carboxylate Sera-Mag Speed Beads (Fischer Scientific) were diluted to 100 μg/μL in 10% formic acid and 5 μL were added to each lyophilized sample. Aggregation of the peptides onto beads was induced via the addition of 195 μL acetonitrile and incubating for 18 min while shaking at 100 rpm on the orbital shaker. Next, the supernatant was removed from the magnetic rack in two steps. The beads were washed 2 times with 180 μL acetonitrile and subsequently dried. In the final steps, the beads were resuspended in 20 μL 0.1% formic acid in water and sonicated in an Ultrasonic Cleaner USC-T (VWR) for 10 min and the supernatant was transferred to a new plate. The purified peptides were dissolved in 0.1% formic acid and used for LC-MS/MS analysis.

## Preparation of proteomics samples without enrichment of newly synthesized proteins

For this, 50 μg of AHA and SILAC pulse-labeled Hela cell lysate and 100 μg of 10-day-long SILAC-labeled cell lysate were used for the preparation of non-enriched proteome samples, which were used to compare non-enriched and enriched NSP samples and to evaluate different mass spectrometry acquisition methods. For the targeted analysis of IFNg target proteins, 50 μg cell lysate was used. Cells were lysed with 1% SDS and 300 mM HEPES (pH 8.0) containing lysis buffer, sonicated with a probe sonicator (Branson) for a total of 1 min per lysate. The protein concentration of the lysates was determined using a BCA assay (Pierce) and the lysates of the labeled Hela cells were combined with 2 defined ratios, to a total of 100 μg protein input. The 2 different SILAC mix samples were created to represent an enriched newly synthesized proteome sample and a pulse-labeled sample

without enrichment. The mix 1 sample consisted of 70% unlabeled protein, 15% heavy and 15% intermediate SILAC labeled protein, whereas the mix 2 sample consisted of 40% heavy, 40% intermediate labeled protein and 20% unlabeled protein. Proteome samples of the labeled lysates and lysates of cells treated with IFNg, for targeted validation experiment via PRM, were prepared using the SP3 protein purification protocol[64]. Aggregation of the proteins on the magnetic beads was induced by the addition of acetonitrile to a final concentration of 50% (v/v) and incubating at room temperature for 18 min. The beads were subsequently washed twice with 80% (v/v) ethanol and acetonitrile. On bead digestion of proteins in the samples was carried out with 1 μg sequencing grade Trypsin (Promega) (protease to protein input ratio 1/50) in 100 mM ammonium bicarbonate (pH 8.0) for 16 h at 37 °C. Following the tryptic digestion, the supernatant was removed from the beads using a magnetic rack and trifluoroacetic acid (TFA) was added to a final concentration of 1% (v/v).

## LC-MS/MS

Quantitative measurements of tryptic peptides, of the enriched newly synthesized proteins and proteomics samples, were carried out using an EASY-nLC 1200 system (Thermo Fischer Scientific) coupled to a QExactive HF mass spectrometer (Thermo Fischer Scientific).

The peptides were separated by reverse-phase liquid chromatography using 0.1% formic acid (solvent A) and 80% acetonitrile (solvent B) as mobile phases. Peptide separation occurred on an Acclaim PepMap trap column (Thermo Fischer Scientific, C18, 20 mm × 100 μm, 5 μm C18 particles, 100 Å pore size) and a nanoEase M/Z peptide BEH C18 analytical column (Waters, 250 mm × 75 μm 1/PK, 130 Å, 1.7 μm). The samples were loaded onto the trap column with the constant flow of solvent A at a maximum pressure of 800 bar. The analytical column was equilibrated with 2 μL solvent A at a maximum pressure of 600 bar heated to 55 °C using a HotSleeve+ column oven (Analytical SALES & SERVICES). The peptides were eluted with a constant flow rate of 300 nL/min. The concentration of solvent B was gradually increased during the elution of the peptides in either of three different HPLC gradients used in this study.

The gradient used for the analysis of the newly synthesized proteome and input samples started with 4% solvent B and was increased to 6% in the first 1 min, increased to 27% at 70 min and further increased to 44% after 85 min. After 85 min the percentage of solvent B was raised to 95%. After 95 min the system was re-equilibrated using 5% solvent B for 10 min. The gradient used for the analysis of the SILAC-labeled Hela benchmark samples started with 3% solvent B for the first 4 min, increased to 8% after 4 min and to 10% after 6 min. After 68 min the percentage of solvent B was raised to 32% and after 86 min to 50%. From 87 min to 94 min of the gradient the percentage of solvent B increased to 100%. After 95 min the system was re-equilibrated using 3% solvent B for 10 min. The gradient used for the analysis of the newly synthesized proteome samples of IFNg-treated Hela cells started with 4% solvent B and was increased to 6% in the first 1 min, increased to 27% at 51 min and further increased to 44% after 70 min. After 70 min the percentage of solvent B was raised to 95%. After 80 min the system was re-equilibrated using 5% solvent B for 10 min. Eluting peptides were ionized and injected into the mass spectrometer, using the Nanospray flex ion source (Thermo Fischer Scientific) and a Sharp Singularity nESI emitter (ID = 20 μm, OD = 365 μm, L = 7 cm, α = 7.5°) (FOSSILIONTECH), connected to a SIMPLE LINK UNO-32 (FOSSILIONTECH). A static spray voltage of 2.5 kV was applied to the emitter and the capillary temperature of the ion transfer tube was set to 275 °C.

The QExactive HF mass spectrometer was operated in the data-dependent (DDA) or data-independent (DIA) mode for the five different acquisition methods evaluated in this study. Detailed descriptions of the different HPLC- and mass spectrometry methods can be found in the Supplementary Methods. DDA methods for the measurements of the NSP samples, input samples and SILAC-labeled Hela samples

only differed in their HPLC gradient. In all cases, a full scan range of 375–1500 m/z, Orbitrap resolution of 60000 FWHM, automatic gain control (AGC) target of 3e6 and maximum injection time of 32 ms was set and data-dependent MSMS spectra were acquired using a Top 20 scheme, using a fixed scan range from 200 to 2000 m/z and fixed first mass of 110 m/z. The quadrupole isolation window was set to 2.0 m/z and normalized collision energy was set to 26. The Orbitrap resolution was set to 15000 FWHM with an AGC target of 1e5 and a maximum injection time of 50 ms. MSMS spectra were acquired in profile mode and a charge state exclusion of 1, 5–8 and >8 was defined. An intensity threshold of 2e4 and a minimum AGC target of 1e3 was set.

Two different DIA methods were used for the analysis of the SILAC-labeled Hela benchmark samples. In DIA method 1 a full scan range of 400–1000 m/z with Orbitrap resolution of 60000 FWHM, 3e6 AGC target and maximum injection time of 20 ms was set. Data-independent MSMS spectra were acquired with an Orbitrap resolution of 30000 FWHM, maximum injection time of 50 ms and AGC target of 1e6, using 26 equally sized, 1 Th overlapping isolation windows with a width of 23.3 m/z. The normalized collision energy for the fragmentation of precursor ions was set to 27 and a fixed first mass of 200 m/z was set for the acquisition of the MSMS spectra. DIA method 2 used three full scans with a range from 400 to 1000 m/z, Orbitrap resolution of 120000 FWHM, 3e6 AGC target and maximum injection time of 20 ms. Data-independent MSMS spectra were acquired with an Orbitrap resolution of 30000 FWHM, maximum injection time of 50 ms and AGC target of 1e6, using 27 equally sized, 1 Th overlapping isolation windows with a width of 23.2 m/z. The normalized collision energy for the fragmentation of precursor ions was set to 27 and a fixed first mass of 200 m/z was set for the acquisition of the MSMS spectra. For the analysis of the IFNg-treated NSP samples, a different DIA method was used. This third DIA method featured a full scan range of 400–1000 m/z, Orbitrap resolution of 60000 FWHM, 3e6 AGC target and maximum injection time of 40 ms. Data-independent MSMS spectra were acquired with an Orbitrap resolution of 30000 FWHM, maximum injection time of 40 ms and AGC target of 1e6, using 28 equally sized, 1 Th overlapping isolation windows with a width of 22.0 m/z. The normalized collision energy for the fragmentation of precursor ions was set to 27 and a fixed first mass of 200 m/z was set for the acquisition of the MSMS spectra. Targeted proteomic analysis of selected IFNg target protein candidates was carried out via parallel reaction monitoring (PRM). The previously described 90 min gradient (identical to DIA method number 3) was applied. Targeted precursors were isolated using 1.4 m/z wide isolation windows and fragmented using various normalized collision energies. The full scan resolution was set to 60000, scan range was set from 370 to 1015 m/z with target AGC of 3e6 and the maximum inject time to 30 ms. Subsequent targeted MS/MS scans were carried out using 1.4 Th wide quadrupole isolation windows, with a maximum inject time of 200 ms and Orbitrap resolution of 60000.

Detailed summaries of the LC-MS methods used in this study can be found in the Supplementary Information.

## Statistics and reproducibility

Statistical analysis of the data was carried out using the R software environment (version 4.0.3) with additional software packages, which are specified in the Data analysis section. No statistical method was used to predetermine the sample size. The number of experimental and technical replicates was based on considerations from previous experiments. Due to low confidence identifications in the PRM analysis, KLF3 precursors were excluded from downstream analysis, otherwise, no data were excluded from the analysis in this study. Information on the statistical tests is described in the figure legends and Data analysis section.

The 96-well plate positions and measurement order for the IFNg-treatment experiment were randomized. To avoid potential carry-over effects in the LC-MS measurements, samples from the dilution series of magnetic alkyne agarose beads and protein input were acquired in ascending order.

## Data analysis

Raw files from DDA measurements were processed using Maxquant (version 2.0.3) and the Andromeda search engine[65], using a human proteome fasta file, retrieved from the SwissProt database (version from February 2021 with 20934 entries). The enzymatic digestion was set to Trypsin/P and a maximum of 2 missed cleavages per peptide were allowed. For the analysis of NSP data, raw files of both the NSP and global proteome samples were processed together, using Maxquant (version 2.0.3). The multiplicity was set to 3, comprising a light channel, an intermediate channel with Arg6 and Lys4 and a heavy channel with Arg10 and Lys8. Cysteine carbamidomethylation was set as fixed modification, whereas Methionine oxidation, N-terminal acetylation, and deamidation of Asparagine and Glutamine were set as variable peptide modifications. The Re-quantify function was enabled, match-between-runs was disabled and other search functions were left with default parameters. Minimum peptide length was set to 7 and max peptide mass was set to 4600 Da. PSM-, Protein- and site decoy fraction FDR were set to 1%. The minimum delta score threshold for unmodified peptides was set to 6, and 40 for modified peptides. Only for the analysis of the SILAC-labeled Hela benchmark samples the match-between-runs function was enabled. Unique and razor peptides were used for quantification and normalized SILAC ratios and iBAQ values were calculated. The minimum ratio count was set to 0 to not exclude identifications in single SILAC channels.

Raw files from DIA measurements were analyzed using DIA-NN[46] (version 1.8.1). A predicted spectral library was generated from the fasta file, which was also used in the Maxquant searches. Additionally, a fasta file containing common protein contaminants was added for the spectral library prediction[66]. Default settings were used for the spectral library prediction, with the addition of Methionine oxidation as variable modification. For the processing of the raw files, the default settings of DIA-NN were used with additional functions from the plexDIA module enabled[44]. Three SILAC channels with mass shifts corresponding to Lys, Lys4 (+4.025107 Da), Lys8 (+8.014199 Da), Arg, Arg6 (+6.020129 Da), Arg10 (+10.008269 Da) and an additional decoy channel with Lysine (+12.0033 Da) and Arginine (+13.9964 Da) were registered. Translation of retention times between peptides within the same elution group was enabled. The first $^{13}C$-isotopic peak and monoisotopic peak was included for the quantification and the MS1 deconvolution level was set to 2. Peptide length range was set from 7 to 30, precursor charge rate was set from 1 to 4, mass of charge (m/z) range of the precursors was set from 300 to 1800 and fragment ion m/z range was set from 200 to 1800. Precursor FDR was set to 1%. Precursor matrix output tables were filtered for FDR < 0.01 and additionally for channel $q$-value < 0.01 and translated $q$-value < 0.01. The MBR function in DIA-NN was enabled, except for the protein input dilution (Supplementary Fig. 6). However, the "first-pass-search" results prior to MBR were used for the SILAC benchmark (Fig. 4), to exclude the effects of MBR for the comparison. For the analysis of the protein input dilution series with plexDIA, the "unrelated runs" option in the DIA-NN GUI was ticked.

The output tables from Maxquant ("ProteinGroups.txt", "evidence.txt") and DIA-NN ("report.pr_matrix_channels_translated.tsv" and "report.pr_matrix_channels_ms1_translated.tsv") were processed in the R software environment (version 4.0.3) using custom scripts. Identified contaminants were removed and protein abundance was calculated using the MaxLFQ algorithm, applied to the individual SILAC channels, using the iq (version 1.9.6) R package function "process_long_format()"[67]. For MS1- and MS2-based quantification, the "Ms1.translated" and "precursor.translated" quantity was used for the MaxLFQ calculation, respectively. Protein-group SILAC ratios were

calculated for each sample using the LFQ values. For the analysis of the IFNg-treated Hela cells, only MS2-based quantification was used. Principle component analysis (PCA) of the log2-transformed SILAC ratios was performed using the "prcomp" function of the stats (version 4.0.3) R package. Peptide hydrophobicity (GRAVY index) was calculated using the "hydrophobicity" function of the Peptides (version 2.4.4) R package. For differential expression analysis, only unique protein groups (single Uniprot identifier) with a minimum of 2 SILAC ratios values in 3 replicates were used. Differential expression tests were carried out using the Limma (version 3.46.0)[68] and DEqMS (version 1.8.0)[69] R/Bioconductor packages, by fitting the data onto a linear model and performing an empirical Bayes moderated $t$-test. The number of precursors, with consideration of modified peptide sequences and charge but not SILAC channels, of each protein group was included as a factor for the variance estimation in DEqMS. Overrepresentation enrichment analysis of significantly deregulated protein groups (absolute log2 fold change > 0.585 and adjusted $p$-value < 0.05), from the IFNg time course experiments, was carried out using a hypergeometric test via the "enricher" function of the clusterProfiler (version 3.18.0)[70] R/Bioconductor package. Gene set enrichment analysis was carried out using the sorted log2 fold change values of all quantified protein groups, using the "GSEA" and "gseGO" function of the clusterProfiler (version 3.18.0)[70] R/Bioconductor package. Gene lists of the Molecular Signatures Database were retrieved and analyzed using the msigdbr (version 7.5.1) R package of the CRAN software repository[71]. Gene sets of the Hallmark (H) subset were included in the analysis. Additionally, the top 500 significantly upregulated genes from the 24 h IFNg-treated Hela cells of the deposited RNA-seq dataset (GSE150196, more specifically "GSE150196_RNA-seq_DESeq2_priming_vs_naive.tab")[47] and the top 500 target genes from a public STAT1 ChIP-seq dataset from 30 min IFNg-treated Hela S3 cells (ENCSR000EZK, more specifically "ENCFF039MZH.bed") were included in the enrichment analysis (Supplementary Data 7)[72]. The ChIP-seq data was processed using a custom R-script, only genes within 2000 bp upstream or downstream of the annotated transcription start sites of the hg38 reference genome were included in the final STAT1 target gene list. Quantified protein groups in the respective condition were included as background gene list for the enrichment and $p$-values were adjusted using the Benjamini–Hochberg approach[73]. The number of datasets in which differential expression of selected proteins or genes was reported in response to IFNg in human cells or tissues, was retrieved from the interferome.org database[48]. Raw PRM data was analyzed using Skyline (version 22.2.0.527)[74]. A predicted spectral library, which was generated by Prosit was used for the analysis in Skyline[75]. Relative quantification and statistical analysis were carried out using the sum of TIC-normalized fragment ion peak areas via MSstats (version 4.2.2.0)[76].

### Reporting summary

Further information on research design is available in the Nature Portfolio Reporting Summary linked to this article.

## Data availability

The mass spectrometry data and processing parameters for the Maxquant, DIA-NN and Skyline search engines have been deposited to the ProteomeXchange Consortium via the PRIDE[77] and Panorama Public[78] repository. The data from the initial method optimizations are available with identifier PXD036886 (semi-automated enrichment of AHA-containing newly synthesized proteins with magnetic alkyne agarose beads), data from the SILAC labeled benchmark samples with identifier PXD039580 (Benchmarking of DDA and plexDIA analysis with SILAC-labeled Hela cell samples) and PXD039578 (Benchmarking plexDIA analysis with SILAC-labeled Hela cell samples), and data from the IFNg time course experiment with identifier PXD038915 (Newly synthesized proteome analysis of interferon-gamma treated Hela cells). Raw files

and data from the PRM analysis are available with ProteomeXchange identifier PXD043967 at Panorama Public (Label-free Parallel Reaction Monitoring (PRM) analysis of selected novel potential targets of IFNg stimulation in Hela cells). Raw files and data from the plexDIA input dilution analysis are available with PRIDE identifier PXD043817 (Applying plexDIA analysis to newly synthesized proteome samples generated using a semi-automated enrichment protocol). Source data are provided with this paper.

## Code availability

R scripts for the processing of DIA-NN and Maxquant output tables and analysis of the proteomic data in this study, together with the protocol files for the semi-automated enrichment protocol (for an Agilent Bravo liquid handling robot) are available from: https://github.com/krijgsveld-lab/QuaNPA[79].

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

## Acknowledgements
The authors would like to thank Dr. Vadim Demichev for helpful comments regarding the analysis of SILAC-labeled samples with DIA-NN. This work was funded in part by the German Ministry of Education and Research (BMBF), as part of the National Research Node "Mass spectrometry in Systems Medicine" (MSCoreSys), under grant agreement 161L0212A.

## Author contributions
T.B. and J.K. conceived the study and designed the approach. T.B. developed the protocol for the preparation of magnetic alkyne agarose beads. T.B. prepared samples, acquired and analyzed data. T.M. and T.B. created the automated enrichment protocol. J.K. together with T.B. wrote the manuscript.

## Funding

## Competing interests
The authors declare no competing interests.
