## [Peer Review File · Nature Communications]

nature portfolio

Peer Review File

An integrated workflow for quantitative analysis of the newly synthesized proteomeReviewer #1 (Remarks to the Author):

In the manuscript entitled: "An integrated workflow for quantitative analysis of the newly synthesized proteome", Bortçen et al. present a QuanPA workflow for the quantitative analysis of the newly synthesized proteome (NSP). Although the importance of protein synthesis is well recognized, quantitative analysis of protein synthesis has proven to be challenging. This workflow includes an optimized semi-automated sample preparation protocol and the application of multiplexing DIA-MS to reduce the sample input, increase the proteomic depth, reduce missing values, and improve quantitative accuracy of the analysis. The authors first optimized an automated protocol including the synthesis of magnetic alkyne agarose beads and NSP enrichment and SP3 peptide purification using a Bravo liquid handling system enabling the analysis of 8-96 samples in parallel with a minimized sample input. Then, two benchmark samples were used to compare different MS methods and data processing software for NSP analysis showing a better performance of DIA MS2-based quantification in plexDIA than DDA. Finally, the optimized protocol was applied to study the response to IFN γ in HeLa cells.

The main novelty of the manuscript lies in the technology improvement enabling parallel processing of multiple samples and using lower sample input. Furthermore, the NSP analysis depth and quantification benefits from a novel combination with a DIA-based MS method. The manuscript is well written and presents high quality data. My main concern is the novelty of the method application part, since a similar analysis of IFN γ signaling has been performed recently (Kleinpenning, 2020). As a benchmark experiment, the analysis of IFN γ signaling showed that QuanPA is able to gain biologically relevant insights by comparing to other omic datasets and annotation databases and lead to identification of potentially novel targets of IFN γ signaling. However, to increase the impact of the biological findings of the current study, I would strongly suggest performing follow-up experiments focusing, e.g., on the validation of some of the potentially novel targets of IFN γ signaling or extending the study to a more interesting system.

Comments:

1) Page 3, line 84-85: "Although DIA is directly compatible with label-free quantification, it has been scarcely applied in combination with SILAC labelling because of challenges in data analysis 41,42,43. Recently, plexDIA was introduced in the DIA-NN environment, demonstrating deep proteome coverage and quantitative accuracy for the analysis of multiplexed samples with non-isobaric labels (mTRAQ). Conceptually, plexDIA could also be applied to SILAC-labelled samples, in particular for the analysis of NSPs."

Another DIA-data analysis software, Spectronaut has been capable of analyzing SILAC-DIA samples for several years, and it has been later optimized for the analysis of pSILAC samples (Salovska et al., 2020) enabling a straightforward analysis of SILAC and pSILAC DIA-MS data. The way this paragraph of the introduction is written gives a false impression that the analysis of such data is problematic and plexDIA offers a unique solution, which is not true.

2) Page 4, paragraph starting at line 132: Please briefly clarify in the text and related figure description how the MS data for this optimization step were acquired and analyzed. It is not clear from the manuscript until the next results section.

3) Furthermore, in Figure 3B-D and 3E-F, it is not clear whether the analysis is comparing the same set of overlapping ids identified across all conditions or every boxplot/violin plot represents all protein ids identified in each condition (which would be fair). For Figure 3C, also please clarify whether these are raw or normalized SILAC ratios. In the Method section (line 597) it is mentioned that the normalized ratios were used. The authors might also consider presenting peptide-precursor level identification and ratios, since collapsing the peptides into protein level H/M ratios might remove a lot of quantitative noise.

4) Page 6, Line 178-180: In the recent paper, plexDIA has been used to analyze multiplexed samples using mTRAQ labeling in which peptide precursor with C-terminal

arginine generate unlabeled y-ions during fragmentation. SILAC labeling presents a similar challenge when using MS2 quantification due to the lack of labeling of the majority of the b-ions and the co-isolation of the labeling pairs in the same DIA window. Since this is important for the SILAC MS2 quantification, could you please briefly clarify how DIA-NN handles these unlabeled b-ions?

5) Figure 4C – in addition to protein level identification it might be also informative to also show the number of identified peptides.

6) Figure 4D-E – Please, clarify whether the presented analysis is based on a set of overlapping protein ids between all the methods used or all ids identified in the respective experiments. It might be informative to evaluate the precision and accuracy for the set of overlapping ids and the “extra” ids identified by DIA and also show the precursor level result when possible. Are the observed differences statistically significant?

Please also add a brief clarification in the methods section how DIA-NN performs the MS1 and MS2-based quantification, is there any fragment ion or precursor ion level filtering? Especially considering the b-ion interference mentioned above?

7) Page 7, line 236-243 (Figure 5A): The depiction of the AHA and SILAC labeling time points in Figure 5A is not very clear, and it is also not described in the relevant section of the manuscript. Perhaps a different visualization using a timeline would be better to directly understand that for the later timepoints, the labeling was only performed in the last 6 hours?

8) Page 8, line 248-249, “Shorter labeling times did not lead to noticeable reduction in the number of quantified proteins or precision, etc.”: While the good identification numbers in the early time points might be explained by the enrichment, the analysis of all samples together can also affect the identification result in individual samples, e.g., due to the MBR function. To make this conclusion, have you also tried to process the samples separately and are the identification results consistent?

9) Page 8, line 258-263, Figure S4, comparison to PhosID. When relating to the PhosID dataset, the supplementary figure shows more differentially regulated proteins after 4 hours in the PhosID dataset compared to QuanNPA. It might be fair to comment on this observation too.

10) Page 8, line 276-277, Figure 6: The two log₂FC cutoffs mentioned in the manuscript and shown in the Figure 6 legend are a bit confusing. I would suggest also adding the $p < 0.05$ and $|\log_2FC| > 1$ cutoff into the Figure 6, for example in the grey box next to the label “differentially expressed proteins”. Moreover, in the legend of Figure 6, it is a bit confusing why all subgroups are depicted with an asterisk while the asterisk indicate known link to the IFN γ treatment as described in the legend text.

11) Page 11, line 364-373: The authors hypothesize that plexDIA might have difficulties with quantifying the “extreme” ratios in mix1 due to the fact that DIA-NN does not offer the re-quantify function that is available in MaxQuant. It would be relevant in this section to discuss the fact that another DIA-data processing software, Spectronaut, offers the “inverted spike-in workflow” that has been shown to recover even more “extreme” ratios of H/L pairs, up to 1:16 in a benchmark sample (Salovska 2020 and 2021).

12) Page 15, line 512 and Page 17, line 590 – What is meant by the “global proteome sample”?

13) Page 17, line 594-597 – It is not clear in which analyses the MBR function was enabled and disabled. Maybe add a link to specific figures here? As I mentioned above, it is also unclear to me in which analyses the raw and normalized SILAC ratios were used, as reported by MaxQuant, and in which the authors used the corresponding light and heavy signals and assembled the proteins using MaxLFQ.

14) Page 26, Figures 5 and 6 legends: Please indicate the test used for the statistical analysis of the results depicted in these figures in the figure legend. Please, also add this information to Figure S4 and S5.

15) Page 27, Figure S5, please add statistical significance of the correlations to the correlation plots.

Reviewer #2 (Remarks to the Author):

In the manuscript "An integrated workflow for quantitative analysis of the newly synthesized proteome", Bortecen et al. present an optimized workflow to study newly synthesized proteins (NSPs) which tackles several challenges such as high sample inputs, lack of automation and sufficient throughput, long sample preparation and measurement times, and limited proteome coverage. To support the validity and benefits of their workflow, they apply it to an already well-studied biological system analyzing NSPs in response to IFNg and compare their results with existing data. The authors present a well-written and insightful manuscript with clear figures. The single sample preparation steps follow previously described protocols, but the authors assemble them in a coherent workflow, which they also describe in detail in the method section and which thus appears easy to follow. Given the downscaling and increased throughput of this optimized workflow, this study will likely raise some interest in the proteomics field. However, it remains mainly a method paper with limited new biological insight.

I believe that addressing the following points will enhance the quality and impact of the presented manuscript further:

1. The authors enrich and quantify the newly synthesized proteome and conclude that regulations of these NSP result from a change in protein synthesis. This appears reasonable given e.g. the biology of their IFNg experiment, but this conclusion can still be misleading.

Allow me the following thought experiment: Let us assume that newly synthesized and preexisting proteins are degraded at identical rates, i.e. the cell cannot distinguish between a rather new and a rather old protein. This is an assumption frequently made in pulsed SILAC/turnover experiments and a prerequisite of curve fitting to time course pulsed SILAC experiments. Further, let us assume that the synthesis rate of protein A remains the same after a treatment, but its degradation rate increases. In this case, a larger fraction of what is synthesized newly will be degraded at any given timepoint and we would observe a reduced amount of newly synthesized protein A. Following the claims in the paper, however, this situation would always be interpreted as a slowed-down synthesis of protein A. Likewise, if the degradation of protein B was slower and its synthesis rate remained the same, the amount of newly synthesized protein B would increase because a smaller fraction is degraded, which would be interpreted falsely as a faster synthesis.

It is a common misconception that pulse experiments can distinguish between synthesis and degradation, although we can only measure the combined effects of both in a pulsed SILAC experiment. I believe this misconception should be addressed. I hope the authors will follow my argument and add a section to their manuscript which should state that, even though they do enrich for newly synthesized proteins, observed changes in NSPs can also come from a change in their degradation.

2. In supplementary figure 2, combined (labeled + unlabeled) intensities are used as a proxy for total protein abundance (i.e. protein copies). While there is some correlation between measured protein intensities and actual abundance, the intensities are also markedly affected by protein length (i.e. number of potentially detectable peptides). Please use iBAQ or similar as a proxy for absolute protein abundances to establish that 'stickiness' does not depend on protein abundance.

3. I was intrigued by the authors' comparison of the two different DIA methods m1 and m2. Do the authors have any suggestions as to why m2 yields consistently more quantified proteins than m1 for both MS1- and MS2-based quantification despite longer cycle times and fewer MS2 scans? This appears counterintuitive given the notion in the DIA field that more MS2 scans and faster cycle times are better.

4. An interesting observation is also that DDA yields >10% fewer quantified proteins when the light channel is not the most abundant one. This may indicate a long-existing bias in how the data is processed in MaxQuant. Perhaps it would be worth explicitly mentioning this in the manuscript.

5. Finally, in line 203 and following, the authors state that "the gained proteins in the DIA data compared to DDA are likely to be in the lower abundance range". I suggest that the authors support this statement with actual analyses. For example, what is the

overlap in proteins between DDA and DIA? Are the proteins exclusive to DIA lower in intensity than the ones shared with DDA?

Minor edits/comments:

1. Please consistently capitalize the first word following the numbering in Figure 1 (e.g. "1. Preparing magnetic...")
2. Please define AFA.
3. Line 255, 266, 277: Closing brackets are missing.
4. Line 391: "lysates from" instead of "lysates form"
5. Supplementary Figure 3 needs to be referenced in the main text.
6. Line 474, 526, 527, 549, 957, and 958: "Thermo Fisher" instead of "Thermo Fischer"

Reviewer #3 (Remarks to the Author):

Bortecen et al. introduce a method called QuaNPA, which stands for quantitative analysis of the newly synthesized proteome, that enables the analysis of newly synthesized proteins (NSPs) in response to cellular perturbations. QuaNPA involves pulse-labeling cells with clickable and stable-isotope-coded amino acids, and uses mass spectrometry and data analysis to quantitatively analyze the NSPs. The study found that QuaNPA provides a powerful approach for large-scale investigation of NSPs, and was able to successfully investigate the time-resolved cellular response to interferon-gamma (IFN γ). The main mass spectrometry related principles used in this manuscript have already been introduced significantly earlier and many of them pioneered by the Krijveld group themselves. Nevertheless, I do think that the authors present very impactful improvement relative to most methods available to study changes in protein production at a global scale and at relatively short timescales. The main advantages are that new kind of magnetic beads are presented with a higher capacity to capture newly produced proteins, an automated experimental workflow to isolate NSPs at high efficiency and also that the authors took advantage of the plexDIA workflow to significantly increase protein coverage without increasing mass spec measurement time. Therefore, this highly optimized workflow to measure protein production changes across many conditions, with a significantly lower sample input amount and at higher coverage and throughput, will be extremely useful to the community (admittedly, not all labs will have a robot available to apply the automated workflow, but many will be able to still use the majority of improvements with standard lab equipment). In addition, the manuscript is generally well written and the experiments very thorough. There are only a few (mostly minor) points that I think the authors should address before acceptance.

1. Generally, some of the optimization/characterization of QuaNPA as shown in Figure 3, should actually be at least partly repeated by doing DIA measurements instead of DDA as DIA does indeed show significantly higher sensitivity and therefore some conclusions drawn from the DDA measurements might not be translated 1:1 to DIA measurements. The most important one is the conclusion the authors draw about the protein input amount that is needed for improving protein coverage. Based on the DDA measurements not more than 100ug total protein input is needed for NSP enrichment as additional amount does not lead to higher coverage (Figure 3F). However, due to the increased sensitivity of DIA relative to DDA this might not hold true for the final QuaNPA approach. Also, for example non-enriched samples show fewer quantified protein groups (Figure 3A) might not be 100% true with DIA. It would be great if this could be assessed by the authors.

2. For nearly all comparisons where sensitivity is assessed also peptide level identification numbers should be provided. Especially for the DDA and DIA comparison as DIA-NN is a bit more "aggressive" about the protein grouping, meaning separating proteins already on less stringent criteria into separate groups than other programs.

3. Figure 4D and 4E: for the CV comparison and log₂ ratio comparison between DIA and DDA, I would suggest to also separately look only at the protein groups that overlap

between all measurements. This is probably a bit fairer as DDA measured proteins are probably on average higher expressed and potentially provide a better SNR.

4. I would suggest that when the authors introduce the data for Figure 3, they explicitly mention that the samples were measured by DDA. This gets clear in the next section when plexDIA is introduced, but nevertheless it would already be helpful at this point in the manuscript.

5. For the IFNg experiment: the authors mention that different labeling times were applied. It would be good to maybe have the details about the length of the labeling pulse already in the main text and not just Materials and Methods.

6. Figure legend 3A and 3B – the legend seems to be swapped relative to the figure. Also, in line 346 the sentence seems to refer to Figure 3A not 3B.

7. I am a bit nitpicking here, but the authors use throughout the manuscript the term "protein translation". This term, although often used, does actually not really make sense – it should be either "mRNA translation" or "protein synthesis/production".

REVIEWER COMMENTS

Reviewer #1 (Remarks to the Author):

In the manuscript entitled: “An integrated workflow for quantitative analysis of the newly synthesized proteome”, Bortecen et al. present a QuaNPA workflow for the quantitative analysis of the newly synthesized proteome (NSP). Although the importance of protein synthesis is well recognized, quantitative analysis of protein synthesis has proven to be challenging. This workflow includes an optimized semi-automated sample preparation protocol and the application of multiplexing DIA-MS to reduce the sample input, increase the proteomic depth, reduce missing values, and improve quantitative accuracy of the analysis. The authors first optimized an automated protocol including the synthesis of magnetic alkyne agarose beads and NSP enrichment and SP3 peptide purification using a Bravo liquid handling system enabling the analysis of 8-96 samples in parallel with a minimized sample input. Then, two benchmark samples were used to compare different MS methods and data processing software for NSP analysis showing a better performance of DIA MS2-based quantification in plexDIA than DDA. Finally, the optimized protocol was applied to study the response to IFN γ in HeLa cells.

The main novelty of the manuscript lies in the technology improvement enabling parallel processing of multiple samples and using lower sample input. Furthermore, the NSP analysis depth and quantification benefits from a novel combination with a DIA-based MS method. The manuscript is well written and presents high quality data. My main concern is the novelty of the method application part, since a similar analysis of IFN γ signaling has been performed recently (Kleinpenning, 2020). As a benchmark experiment, the analysis of IFN γ signaling showed that QuaNPA is able to gain biologically relevant insights by comparing to other omic datasets and annotation databases and lead to identification of potentially novel targets of IFN γ signaling. However, to increase the impact of the biological findings of the current study, I would strongly suggest performing follow-up experiments focusing, e.g., on the validation of some of the potentially novel targets of IFN γ signaling or extending the study to a more interesting system.

We thank the reviewer for the assessment and comments. Indeed, the main focus of the manuscript lies on introducing and combining the novel technologies and methods. In fact, we applied QuaNPA to IFN γ stimulation because it is such a well-characterized system. To follow the suggestions of the reviewer we have performed a targeted proteomic analysis via label-free parallel reaction monitoring (PRM), to validate whether the observed changes in NSP levels of the potential novel targets of IFN γ lead to changes in protein abundance upon IFN γ stimulation. Using PRM analysis, we were able to confirm differential expression of novel targets BICD2 and SIN3B, following IFN γ treatment. Interestingly, and in line with our NSP data (Fig 5E and 6A), IFN γ treatment led to an increase in the abundance of SIN3B, while BICD2 was decreased, indicating that QuaNPA can detect expression changes in both directions. Furthermore, we could highlight the quantitative analysis of NSPs is able to measure significant upregulation of canonical IFN γ targets such as ICAM1, TAP1 and STAT1 \leq 4h, whereas increase in overall protein abundance of these 3 proteins as determined by PRM only reached significance at later time points. These data are included in the revised manuscripts as new Figures 6b and Supplementary Figure 9.

Comments:

1) Page 3, line 84-85: "Although DIA is directly compatible with label-free quantification, it has been scarcely applied in combination with SILAC labelling because of challenges in data analysis 41,42,43. Recently, plexDIA was introduced in the DIA-NN environment, demonstrating deep proteome coverage and quantitative accuracy for the analysis of multiplexed samples with non-isobaric labels (mTRAQ). Conceptually, plexDIA could also be applied to SILAC-labelled samples, in particular for the analysis of NSPs."

Another DIA-data analysis software, Spectronaut has been capable of analyzing SILAC-DIA samples for several years, and it has been later optimized for the analysis of pSILAC samples (Salovska et al., 2020) enabling a straightforward analysis of SILAC and pSILAC DIA-MS data. The way this paragraph of the introduction is written gives a false impression that the analysis of such data is problematic and plexDIA offers a unique solution, which is not true.

We thank the reviewer for the comment. Indeed, Spectronaut does enable the analysis of DIA data with SILAC labels. However, we believe that our approach is the first to benchmark the approach, showing that SILAC DIA analysis not only resulted in an increase of protein identifications, but also in quantification accuracy that is comparable to DDA. The study by Salovska et al does not include a benchmark with known SILAC ratios to assess the accuracy of the SILAC ratio quantification by Spectronaut, instead carrying out a comparison of MS1 and MS2-based quantification in which they comment on the reduced standard deviation of the MS2-based ratios in pulse SILAC samples with 3 different labelling times. In our hands, data processed with Spectronaut (version 15) resulted in relatively noisy quantification with many outlier ratios that strongly deviate from the known SILAC mix ratios. Yet, we have changed the paragraph to clarify that the use of plexDIA is not the first or only approach for DIA analysis of SILAC samples.

2) Page 4, paragraph starting at line 132: Please briefly clarify in the text and related figure description how the MS data for this optimization step were acquired and analyzed. It is not clear from the manuscript until the next results section.

We thank the reviewer for pointing out the need to clarify the description. The respective paragraph has been edited to include this information.

3) Furthermore, in Figure 3B-D and 3E-F, it is not clear whether the analysis is comparing the same set of overlapping ids identified across all conditions or every boxplot/violin plot represents all protein ids identified in each condition (which would be fair). For Figure 3C, also please clarify whether these are raw or normalized SILAC ratios. In the Method section (line 597) it is mentioned that the normalized ratios were used. The authors might also consider presenting peptide-precursor level identification and ratios, since collapsing the peptides into protein level H/M ratios might remove a lot of quantitative noise.

We thank the reviewer for pointing out the need to clarify the description. Normalized SILAC ratios have been used, indeed as indicated in the Methods section. It is our understanding that MaxQuant performs median normalization of all precursor SILAC ratios for the protein group ratio, thus we do not expect any different results from aggregation of the peptide level data. Indeed, analysis at the precursor level (Review Figure 1) delivers nearly identical results as those at the protein level (Figure 4D, E). Upon suggestion from

the reviewers, we have included precursor level data of the quantitative SILAC benchmark in Supplementary figure 4.

Review Figure 1. Panel of quantitative metrics of precursor SILAC ratios. MS1 level quantification data is indicated in blue, MS2 level data is indicated in red. A) Coefficient of variation (CV) of the precursor SILAC ratios. B) Distribution of the precursor SILAC ratios and their median deviation from the expected ratios. Upper and lower whiskers extend from the hinges to the highest or lowest values that are within 1.5x the interquartile range. Values outside this range are plotted as dots and represent outliers. Data are based on 3 technical replicates of the single SILAC mix samples. Data are based on 3 technical replicates.

4) Page 6, Line 178-180: In the recent paper, plexDIA has been used to analyze multiplexed samples using mTRAQ labeling in which peptide precursor with C-terminal arginine generate unlabeled y-ions during fragmentation. SILAC labeling presents a similar challenge when using MS2 quantification due to the lack of labeling of the majority of the b-ions and the co-isolation of the labeling pairs in the same DIA window. Since this is important for the SILAC MS2 quantification, could you please briefly clarify how DIA-NN handles these unlabeled b-ions?

We thank the reviewer for pointing out the need to further clarify the requirements for SILAC identification and quantification by DIA-NN. The DIA-NN software, according to its documentation and comments by the developer, uses all fragment ion information for the initial identification of the best scoring labelled precursor. Precursors with the remaining SILAC labels are extracted and scored using a target-decoy approach via a decoy channel (default Lys+13 and Arg+14). Quantification is carried out by initially integrating unique fragment ion peak areas of the best scoring labelled precursor. For the remaining channels, the apex ratio of the unique fragment ions is determined and used to calculate the

normalized (“translated”) quantity of the precursors in the remaining channel. In the case of SILAC labelling, by default only γ -ions are used for quantification.

5) Figure 4C – in addition to protein level identification it might be also informative to also show the number of identified peptides.

We thank the reviewer for the helpful suggestion. A plot of the precursor identifications across the different methods and respective SILAC channels (Review Figure 2) has been added to Supplementary figure 5.

Review figure 2. A) Number of identified precursors across the 3 SILAC channels in the benchmark data. MS1 level quantification data is indicated in blue, MS2 level data is indicated in red. B) Number of precursor SILAC ratios in the benchmark data. Data are based on 3 technical replicates.

6) Figure 4D-E – Please, clarify whether the presented analysis is based on a set of overlapping protein ids between all the methods used or all ids identified in the respective experiments. It might be informative to evaluate the precision and accuracy for the set of overlapping ids and the “extra” ids identified by DIA and also show the precursor level result when possible. Are the observed differences statistically significant?

We thank the reviewer for this suggestion. Figure 4 in the manuscript includes data for all IDs, i.e., not only the shared ~2400 protein groups. Review Figure 3 contains the quantitative metrics of the data subset of proteins identified in both DDA and DIA methods. Review Figure 4 in contrast contains the quantitative

comparison for the subset of protein groups that are not shared between DDA and DIA data (~500-3500). Quantitative precision and accuracy is excellent for the subset of shared protein Ids (Review Figure 3), but is only slightly worse for the subset of unique identifications (between DDA and DIA methods), in case of the plexDIA data. However, unique identification in the DDA data shows visibly worse quantitative accuracy and precision (Review Figure 4).

We therefore conclude that the quantification accuracy and precision of SILAC plexDIA data is high for all quantified protein groups and not just for a subset of protein groups with shared identifications in DDA data.

Review Figure 3. Quantitative metrics of the different methods, limited to a shared subset of approximately 2400 proteins. MS1 level quantification data is indicated in blue, MS2 level data is indicated in red. A) Coefficient of variation (CV) of the precursor SILAC ratios. B) Distribution of the precursor SILAC ratios and their median deviation from the expected ratios. Upper and lower whiskers extend from the hinges to the highest or lowest values that are within 1.5x the interquartile range. Values outside this range are plotted as dots and represent outliers. Data are based on 3 technical replicates of the single SILAC mix samples. Data are based on 3 technical replicates.

Review Figure 4. Quantitative metrics of the different methods, limited to the unique identification subset (for either DDA and DIA methods) of between 500-3500 proteins. MS1 level quantification data is indicated in blue, MS2 level data is indicated in red. A) Coefficient of variation (CV) of the precursor SILAC ratios. B) Distribution of the precursor SILAC ratios and their median deviation from the expected ratios. Upper and lower whiskers extend from the hinges to the highest or lowest values that are within 1.5x the interquartile range. Values outside this range are plotted as dots and represent outliers. Data are based on 3 technical replicates of the single SILAC mix samples. Data are based on 3 technical replicates.

Please also add a brief clarification in the methods section how DIA-NN performs the MS1 and MS2-based quantification, is there any fragment ion or precursor ion level filtering? Especially considering the b-ion interference mentioned above?

According to public documentation on the GitHub page of DIA-NN, only y-ions are used for MS2 based quantification. As far as we know, normalisation in MS1-based quantification is carried out through the same “translated quantity” calculation based on precursor intensities. Since we are users and not developers of the software, we do not know the details of the algorithms in DIA-NN beyond the documentation. We therefore performed the quantitative benchmark to evaluate the capabilities of plexDIA for SILAC.

7) Page 7, line 236-243 (Figure 5A): The depiction of the AHA and SILAC labeling time points in Figure 5A is not very clear, and it is also not described in the relevant section of the manuscript. Perhaps a different

visualization using a timeline would be better to directly understand that for the later timepoints, the labeling was only performed in the last 6 hours?

We thank the reviewer for pointing out the need to improve the experimental scheme for the IFNg treatment time course. A modified visualization has been added to figure 5A.

Review Figure 5. Schematic representation of the experimental design and analysis workflow. The IFNg treatment time is indicated using text labels and the length of the coloured bars, the pulsed SILAC and AHA metabolic labelling time is indicated with red coloration of the bars.

8) Page 8, line 248-249, “Shorter labeling times did not lead to noticeable reduction in the number of quantified proteins or precision, etc..”: While the good identification numbers in the early time points might be explained by the enrichment, the analysis of all samples together can also affect the identification result in individual samples, e.g., due to the MBR function. To make this conclusion, have you also tried to process the samples separately and are the identification results consistent?

We thank the reviewer for commenting on the influence of the match between runs (MBR) function. In DIA-NN MBR is carried out through a second pass search with a modified spectral library, which is refined with confident identifications across all analysed samples. To compare the results without MBR, the output tables from the first pass search (prior to MBR) can be used for subsequent analysis. As highlighted in the figure panel below, the results are not significantly affected by MBR, except the approximately 300 fewer quantified protein groups across all time points.

Review Figure 6. Number of quantified protein groups in the samples with the indicated IFNg treatment time points. Numbers indicate the average of three replicates (indicated individually with black dots). C) Boxplots indicating the coefficient of Variation (CV) values of the quantified protein groups for each time point. Upper and lower whiskers extend from the hinges to the highest or lowest values that are within

1.5x the interquartile range. Values outside this range are plotted as dots and represent outliers. D) Principal component analysis (PCA) of the NSP samples.

9) Page 8, line 258-263, Figure S4, comparison to PhosID. When relating to the PhosID dataset, the supplementary figure shows more differentially regulated proteins after 4 hours in the PhosID dataset compared to QuanNPA. It might be fair to comment on this observation too.

We thank the reviewer for the comment. Indeed, a larger number of differentially regulated proteins is reported in the 4 h timepoint of the PhosID dataset. We have mentioned this fact in the edited section of the manuscript, as follows:

“However, fewer differentially regulated proteins were detected at the 4 h IFN γ treatment time point, compared to data produced with the PhosID workflow.”

10) Page 8, line 276-277, Figure 6: The two log₂FC cutoffs mentioned in the manuscript and shown in the Figure 6 legend are a bit confusing. I would suggest also adding the $p < 0.05$ and $|\log_2\text{FC}| > 1$ cutoff into the Figure 6, for example in the grey box next to the label “differentially expressed proteins”. Moreover, in the legend of Figure 6, it is a bit confusing why all subgroups are depicted with an asterisk while the asterisk indicate known link to the IFN γ treatment as described in the legend text.

We thank the reviewer for the helpful suggestions, and for spotting the error with the asterisk. We have changed Figure 6 accordingly.

11) Page 11, line 364-373: The authors hypothesize that plexDIA might have difficulties with quantifying the “extreme” ratios in mix1 due to the fact that DIA-NN does not offer the re-quantify function that is available in MaxQuant. It would be relevant in this section to discuss the fact that another DIA-data processing software, Spectronaut, offers the “inverted spike-in workflow” that has been shown to recover even more “extreme” ratios of H/L pairs, up to 1:16 in a benchmark sample (Salovska 2020 and 2021).

We thank the reviewer for referencing the “inverted spike-in” workflow in Spectronaut. We have included a reference to this proposed solution in the manuscript.

12) Page 15, line 512 and Page 17, line 590 – What is meant by the “global proteome sample”?

We thank the reviewer for pointing out the unclear term. The section of the manuscript has been edited and we now prefer to describe the samples as “proteomics samples without NSP enrichment”.

13) Page 17, line 594-597 – It is not clear in which analyses the MBR function was enabled and disabled. Maybe add a link to specific figures here? As I mentioned above, it is also unclear to me in which analyses the raw and normalized SILAC ratios were used, as reported by MaxQuant, and in which the authors used the corresponding light and heavy signals and assembled the proteins using MaxLFQ.

We thank the reviewer for pointing out the need to add further clarifications. Individual links to the respective figures and sections have been added.

For all DDA data, except for the benchmark data (in which unnormalized ratios were compared for all methods), only normalized SILAC ratios as produced by Maxquant are used. For the analysis of DIA data, we apply MaxLFQ normalization across the different SILAC channels, using the "translated quantities" produced by DIA-NN. MaxLFQ normalization was also applied in the original plexDIA publication, but in general produces similar results, to simply averaging precursor SILAC ratios.

14) Page 26, Figures 5 and 6 legends: Please indicate the test used for the statistical analysis of the results depicted in these figures in the figure legend. Please, also add this information to Figure S4 and S5.

A modified empirical Bayes moderated t-test, adjusting t-statistic and p-values with precursor counts, was carried out with the "spectraCounteBayes" function of the DEqMS R package. This has now been clarified in the figure legends and in the methods section.

15) Page 27, Figure S5, please add statistical significance of the correlations to the correlation plots.

We have now added p-values to the Pearson correlation coefficient values in the scatter plots.

Reviewer #2 (Remarks to the Author):

In the manuscript “An integrated workflow for quantitative analysis of the newly synthesized proteome”, Borteçen et al. present an optimized workflow to study newly synthesized proteins (NSPs) which tackles several challenges such as high sample inputs, lack of automation and sufficient throughput, long sample preparation and measurement times, and limited proteome coverage. To support the validity and benefits of their workflow, they apply it to an already well-studied biological system analyzing NSPs in response to IFN γ and compare their results with existing data. The authors present a well-written and insightful manuscript with clear figures. The single sample preparation steps follow previously described protocols, but the authors assemble them in a coherent workflow, which they also describe in detail in the method section and which thus appears easy to follow. Given the downscaling and increased throughput of this optimized workflow, this study will likely raise some interest in the proteomics field. However, it remains mainly a method paper with limited new biological insight.

I believe that addressing the following points will enhance the quality and impact of the presented manuscript further:

1. The authors enrich and quantify the newly synthesized proteome and conclude that regulations of these NSP result from a change in protein synthesis. This appears reasonable given e.g. the biology of their IFN γ experiment, but this conclusion can still be misleading. Allow me the following thought experiment: Let us assume that newly synthesized and preexisting proteins are degraded at identical rates, i.e. the cell cannot distinguish between a rather new and a rather old protein. This is an assumption frequently made in pulsed SILAC/turnover experiments and a prerequisite of curve fitting to time course pulsed SILAC experiments. Further, let us assume that the synthesis rate of protein A remains the same after a treatment, but its degradation rate increases. In this case, a larger fraction of what is synthesized newly will be degraded at any given timepoint and we would observe a reduced amount of newly synthesized protein A. Following the claims in the paper, however, this situation would always be interpreted as a slowed-down synthesis of protein A. Likewise, if the degradation of protein B was slower and its synthesis rate remained the same, the amount of newly synthesized protein B would increase because a smaller fraction is degraded, which would be interpreted falsely as a faster synthesis. It is a common misconception that pulse experiments can distinguish between synthesis and degradation, although we can only measure the combined effects of both in a pulsed SILAC experiment. I believe this misconception should be addressed. I hope the authors will follow my argument and add a section to their manuscript which should state that, even though they do enrich for newly synthesized proteins, observed changes in NSPs can also come from a change in their degradation.

We thank the reviewer for the positive assessment and for insightful comments.

We fully agree with the reviewers position and have now included a paragraph that mentions the influence of protein degradation .

2. In supplementary figure 2, combined (labeled + unlabeled) intensities are used as a proxy for total protein abundance (i.e. protein copies). While there is some correlation between measured protein intensities and actual abundance, the intensities are also markedly affected by protein length (i.e. number

of potentially detectable peptides). Please use iBAQ or similar as a proxy for absolute protein abundances to establish that 'stickiness' does not depend on protein abundance.

We thank the reviewer for the suggestion, we have now repeated the analysis by using iBAQ values instead of summed intensities (Review Figure 7, now included in the manuscript as Suppl Figure 2A). Interestingly, the results do not differ from the previous comparison, and the highest ratios of labelled NSP over unlabelled previously synthesized proteins are found in highly abundant proteins, in the enriched sample. In contrast to the enriched sample, highly abundant proteins have low ratios of NSP to previously synthesized proteins. This could also be influenced by the use of the re-quantify function, which calculates SILAC ratios with consideration of background intensities. In the case of low abundant proteins/peptides, low intensities and thus poor/low signal to noise ratios lead to low SILAC ratios.

Review Figure 7. Protein metrics of pulse-labeled samples analysed by mass spectrometry with and without prior enrichment of NSPs. A) Ratios of labelled NSPs over unlabelled pre-existing proteins, of enriched newly synthesized proteome samples and samples prepared without NSP enrichment plotted against iBAQ values of the respective protein group.

Furthermore, the classification of all unlabelled proteins as pre-existing proteins is a simplification, since even when assuming perfect depletion of intracellular reserves of Methionine, Lysine and Arginine, protein degradation and amino acid recycling can lead to incorporation of Methionine, and unlabelled Lys/Arg. Thus, not only stickiness is a likely explanation for the detection of unlabelled peptides/proteins. However, we believe our observations illustrate the need for SILAC labels in this and similar methodologies, to confidently distinguish NSPs from previously synthesized sticky proteins, or proteins that have incorporated unlabelled amino acids, which cannot be distinguished between the two mixed conditions.

3. I was intrigued by the authors' comparison of the two different DIA methods m1 and m2. Do the authors have any suggestions as to why m2 yields consistently more quantified proteins than m1 for both MS1- and MS2-based quantification despite longer cycle times and fewer MS2 scans? This appears counterintuitive given the notion in the DIA field that more MS2 scans and faster cycle times are better.

We thank the reviewer for the question. Although both DIA methods m1 and m2 cover the same precursor range (400-1000 m/z), they differ in the number of (equally sized) isolation windows. Method m2 includes

27 isolation windows, whereas m1 includes 26 isolation windows. The small differences in the number of quantified proteins are most likely due to the minor decrease of precursors per fragment spectra, which enables more efficient matching of precursors in DIA-NN. However, the observed differences are very small, since the difference in isolation window size is 0.85 Th. However, for accurate quantification short cycle times are crucial. In case of DIA m2 the MS1 cycle time is approximately 1s and about 3s for MS2.

4. An interesting observation is also that DDA yields >10% fewer quantified proteins when the light channel is not the most abundant one. This may indicate a long-existing bias in how the data is processed in MaxQuant. Perhaps it would be worth explicitly mentioning this in the manuscript.

We thank the reviewer for this suggestion. We are not aware of such biases in Maxquant, but have added a comment to the manuscript accordingly, as follows: “More proteins were quantified in the samples of mix 1, which was primarily composed of unlabelled light protein”.

5. Finally, in line 203 and following, the authors state that “the gained proteins in the DIA data compared to DDA are likely to be in the lower abundance range”. I suggest that the authors support this statement with actual analyses. For example, what is the overlap in proteins between DDA and DIA? Are the proteins exclusive to DIA lower in intensity than the ones shared with DDA?

We thank the reviewer for the suggestion. We have now prepared a figure (Review Figure 8), in which we plot the ranked intensities of proteins, identified in both or either the DIA or DDA data (for SILAC mix 2). Indeed, the intensity of proteins that were only identified using DIA is lower than for the overlap or exclusive DDA identifications. Due to the different protein grouping of the software, we only included unique protein groups for the comparison, to enable merging of the data. We have also added the figure to the revised manuscript as Suppl. Figure 4.

Review Figure 8. Comparison of the summed MS1 intensities in the DDA and DIA data of the SILAC benchmark data. Overlaps and unique protein identifications between the different methods and their intensity distributions are indicated.

Minor edits/comments:

1. Please consistently capitalize the first word following the numbering in Figure 1 (e.g. "1. Preparing magnetic...")
2. Please define AFA.
3. Line 255, 266, 277: Closing brackets are missing.
4. Line 391: "lysates from" instead of "lysates form"
5. Supplementary Figure 3 needs to be referenced in the main text.
6. Line 474, 526, 527, 549, 957, and 958: "Thermo Fisher" instead of "Thermo Fischer"

We thank the reviewer for pointing out these errors, which we have corrected in the revised manuscript.

Reviewer #3 (Remarks to the Author):

Bortecen et al. introduce a method called QuaNPA, which stands for quantitative analysis of the newly synthesized proteome, that enables the analysis of newly synthesized proteins (NSPs) in response to cellular perturbations. QuaNPA involves pulse-labeling cells with clickable and stable-isotope-coded amino acids, and uses mass spectrometry and data analysis to quantitatively analyze the NSPs. The study found that QuaNPA provides a powerful approach for large-scale investigation of NSPs, and was able to successfully investigate the time-resolved cellular response to interferon-gamma (IFN γ). The main mass spectrometry related principles used in this manuscript have already been introduced significantly earlier and many of them pioneered by the Krijveld group themselves. Nevertheless, I do think that the authors present very impactful improvement relative to most methods available to study changes in protein production at a global scale and at relatively short timescales. The main advantages are that new kind of magnetic beads are presented with a higher capacity to capture newly produced proteins, an automated experimental workflow to isolate NSPs at high efficiency and also that the authors took advantage of the plexDIA workflow to significantly increase protein coverage without increasing mass spec measurement time. Therefore, this highly optimized workflow to measure protein production changes across many conditions, with a significantly lower sample input amount and at higher coverage and throughput, will be extremely useful to the community (admittedly, not all labs will have a robot available to apply the automated workflow, but many will be able to still use the majority of improvements with standard lab equipment). In addition, the manuscript is generally well written and the experiments very thorough. There are only a few (mostly minor) points that I think the authors should address before acceptance.

We thank the reviewer for the assessment and comments.

1. Generally, some of the optimization/characterization of QuaNPA as shown in Figure 3, should actually be at least partly repeated by doing DIA measurements instead of DDA as DIA does indeed show significantly higher sensitivity and therefore some conclusions drawn from the DDA measurements might not be translated 1:1 to DIA measurements. The most important one is the conclusion the authors draw about the protein input amount that is needed for improving protein coverage. Based on the DDA measurements not more than 100 μ g total protein input is needed for NSP enrichment as additional amount does not lead to higher coverage (Figure 3F). However, due to the increased sensitivity of DIA relative to DDA this might not hold true for the final QuaNPA approach. Also, for example non-enriched samples show fewer quantified protein groups (Figure 3A) might not be 100% true with DIA. It would be great if this could be assessed by the authors.

We thank the reviewer for the suggestion. We have repeated the protein input dilution series experiment using a DIA method (Review Figure 9). This figure is now included as Suppl. Figure 6 of the revised manuscript. Indeed, the greater sensitivity of plexDIA allows for relatively high numbers of identifications, even with 25 μ g protein input for the enrichment of samples. However, similarly to the DDA data, no major gains in the number of quantified proteins and identified precursors is noticeable beyond 100 μ g protein input. Indeed, as the reviewer suggests, our DIA benchmark data (Figure 4) show that samples resembling non-enriched samples do not result in an increase in protein identifications to the same extent as samples that mostly contain labelled proteins (i.e. NSPs).

Review Figure 9. Applying plexDIA for the analysis of samples prepared with the semi-automated NSP enrichment and assessing the ideal protein input range. A) Intensity ratios of samples prepared with differing amounts of protein input. Heavy- and intermediate SILAC labelled precursors (originating from newly synthesized proteins), over light precursor ions (originating from pre-existing proteins). The upper and lower whiskers, of the ratio boxplots, extend from the hinges to the highest or lowest values that are within 1.5x the interquartile range. B) Number of quantified protein groups in the NSP samples prepared with differing protein input amounts. Numbers indicate the average of 2 replicates (red dots). C) Number of identified precursors with light- (L), intermediate- (M) and heavy (H) SILAC labels, in enriched samples with different amounts of protein input, analysed using plexDIA. Data are based on 2 technical replicates, except the 200 µg sample which consists of a single replicate.

2. For nearly all comparisons where sensitivity is assessed also peptide level identification numbers should be provided. Especially for the DDA and DIA comparison as DIA-NN is a bit more “aggressive” about the protein grouping, meaning separating proteins already on less stringent criteria into separate groups than other programs.

We thank the reviewer for the suggestion and have added the number of precursors in the Supplementary figures S2C D, S5A B and S6C.

3. Figure 4D and 4E: for the CV comparison and log2 ratio comparison between DIA and DDA, I would suggest to also separately look only at the protein groups that overlap between all measurements. This is probably a bit fairer as DDA measured proteins are probably on average higher expressed and potentially provide a better SNR.

We thank the reviewer for the suggestion, however, we believe that it is important to consider the full set of quantified proteins for the comparison. The relatively small overlapping subset of 2400 proteins shows high quantitative accuracy and precision for both DIA methods (Review Figure 3, in response to a similar question by Reviewer 1), which indeed look slightly better than the results shown for the entire data set (Figure 4), or the method specific identification subset (Review Figure 4). Yet, we believe that quantitative accuracy for the full data is highly acceptable, thereby making the point that proteins can be accurately quantified across the abundance range.

4. I would suggest that when the authors introduce the data for Figure 3, they explicitly mention that the samples were measured by DDA. This gets clear in the next section when plexDIA is introduced, but nevertheless it would already be helpful at this point in the manuscript.

We thank the reviewer for the suggestion, we have now indicated in the legend of Figure 3 accordingly.

5. For the IFN γ experiment: the authors mention that different labeling times were applied. It would be good to maybe have the details about the length of the labeling pulse already in the main text and not just Materials and Methods.

We thank the reviewer for the suggestion, we have now edited the schematic figure to indicate the differences in the labelling time more clearly.

Review Figure 10. Schematic representation of the experimental design and analysis workflow. The IFN γ treatment time is indicated using text labels and the length of the coloured bars, the pulsed SILAC and AHA metabolic labelling time is indicated with red coloration of the bars.

6. Figure legend 3A and 3B – the legend seems to be swapped relative to the figure. Also, in line 346 the sentence seems to refer to Figure 3A not 3B.

We thank the reviewer for pointing out this mistake. The figure legend and reference to the figure have been corrected.

7. I am a bit nitpicking here, but the authors use throughout the manuscript the term “protein translation”. This term, although often used, does actually not really make sense – it should be either “mRNA translation” or “protein synthesis/production”.

We thank the reviewer for the comment and agree on the importance of precise language. We have changed the sentence to mRNA translation.

Reviewer #1 (Remarks to the Author):

I would like to thank the authors for addressing all my comments and suggestions sufficiently. However, I still have a minor point of disagreement with the following claims in the response letter.

"However, we believe that our approach is the first to benchmark the approach, showing that SILAC DIA analysis not only resulted in an increase of protein identifications, but also in quantification accuracy that is comparable to DDA. The study by Salovska et al does not include a benchmark with known SILAC ratios to assess the accuracy of the SILAC ratio quantification by Spectronaut, instead carrying out a comparison of MS1 and MS2-based quantification in which they comment on the reduced standard deviation of the MS2-based ratios in pulse SILAC samples with 3 different labelling times. In our hands, data processed with Spectronaut (version 15) resulted in relatively noisy quantification with many outlier ratios that strongly deviate from the known SILAC mix ratios."

While I acknowledge that the presented manuscript performed the DIA to DDA benchmark in a very vigorous way, it is not true that this manuscript is the first one to do so. A previously published study by Pino et al., 2021 already showed that DIA was better in quantitative accuracy compared to DDA, so this specific argument does not really support the novelty of the data analysis approach.

Furthermore, I agree it has been shown that some of the recent versions of Spectronaut do provide a less accurate quantification than DIA-NN (Lou et al, Nat Comm, 2023), but despite imperfections, the software has been used to provide important biological insights in several pulsed SILAC studies. Thus, this argument is also not the best supporting the novelty of the approach and why the previous ones should not be specifically mentioned.

Reviewer #3 (Remarks to the Author):

The authors have fully addressed all my concerns and I fully support publication of this manuscript.

REVIEWER COMMENTS

Revision #2:

Reviewer #1 (Remarks to the Author):

I would like to thank the authors for addressing all my comments and suggestions sufficiently. However, I still have a minor point of disagreement with the following claims in the response letter.

“However, we believe that our approach is the first to benchmark the approach, showing that SILAC DIA analysis not only resulted in an increase of protein identifications, but also in quantification accuracy that is comparable to DDA. The study by Salovska et al does not include a benchmark with known SILAC ratios to assess the accuracy of the SILAC ratio quantification by Spectronaut, instead carrying out a comparison of MS1 and MS2-based quantification in which they comment on the reduced standard deviation of the MS2-based ratios in pulse SILAC samples with 3 different labelling times. In our hands, data processed with Spectronaut (version 15) resulted in relatively noisy quantification with many outlier ratios that strongly deviate from the known SILAC mix ratios.”

While I acknowledge that the presented manuscript performed the DIA to DDA benchmark in a very vigorous way, it is not true that this manuscript is the first one to do so. A previously published study by Pino et al., 2021 already showed that DIA was better in quantitative accuracy compared to DDA, so this specific argument does not really support the novelty of the data analysis approach.

Furthermore, I agree it has been shown that some of the recent versions of Spectronaut do provide a less accurate quantification than DIA-NN (Lou et al, Nat Comm, 2023), but despite imperfections, the software has been used to provide important biological insights in several pulsed SILAC studies. Thus, this argument is also not the best supporting the novelty of the approach and why the previous ones should not be specifically mentioned.

We thank the reviewer for pointing out the inaccurate phrasing in the first response letter. Indeed, improved quantification in terms of dynamic range was shown in the paper by Pino et al., 2021. We did not mean to claim that our benchmark, and the use of the plexDIA features of DIA-NN, marks the first case of improved SILAC quantification accuracy using DIA data. However, we believe that it is the first reported case of SILAC DIA data with high quantification accuracy, precision and significantly increased peptide/protein identification.

As the reviewer indicates, this concerned a statement in the response letter, which we hopefully addressed now. Yet, in the main text we verified we did not make such claims. Therefore, we believe that the sections describing and discussing SILAC plexDIA in the main text of the manuscript do not give the impression that we claim novelty in regard to improved quantification accuracy using SILAC DIA analysis. This includes citation of papers using other software tools for analysis of DIA SILAC data.

We fully agree with the reviewer, that Spectronaut is a very powerful and useful software, which has helped with numerous important biological findings, including the pSILAC-based studies. We did not mean to include a software comparison in this paper, but are keen on exploring future tests with more recent versions of the software.

Reviewer #3 (Remarks to the Author):

The authors have fully addressed all my concerns and I fully support publication of this manuscript.

We thank the reviewer for the constructive and helpful comments, which helped us to improve the manuscript.